# The Pulvinus Is the Weak Point for Stem Lodging Resistance in Ripe Barley

**DOI:** 10.3390/plants13223172

**Published:** 2024-11-12

**Authors:** Alberto Gianinetti, Marina Baronchelli

**Affiliations:** Council for Agricultural Research and Economics (CREA), Research Centre for Genomics and Bioinformatics, via S. Protaso 302, 29017 Fiorenzuola d’Arda, PC, Italy; marina.baronchelli@crea.gov.it

**Keywords:** stem lodging, three-point bending test, flexural rigidity, material strength, stem bending strength, stem moisture content, loading point, pulvinus, ripe barley

## Abstract

Stem lodging is a serious problem for the ripe barley crop because it can reduce grain yield and quality. Although biometrical traits (stem diameter and wall thickness) and mechanical properties (stiffness and strength of the culm) have an obvious role in determining lodging resistance, they have only a partial capability to predict lodging resistance. We, therefore, investigated how factors like stem wetting and the point of application of the bending force affect the assessment of these traits. A three-point bending test using a height gauge can provide measures of bending strength (*B_S_*), material strength (σ_b_), modulus of elasticity (*E*), and stiffness (*EI*). Since the first two parameters are of greatest interest, a quick manual method for measuring them is proposed. We used it specifically to compare the results of tests made by loading the bending force either on the node or the internode. It was shown that the pulvinus (which forms a complex with the node) is the weak point for mechanical resistance to bending in ripe barley stems, as a drop in *B_S_* between −31% and −41% (depending on whether the stems were dry or wet) was observed when the loading force was applied on the node/pulvinus complex with respect to the internode. We also found that, overall, *B_S_* plummeted −62% with respect to dry stems when the stems were wetted. This was due to an equivalent (−62%) plunge in σ_b_. Similar drops in *B_S_* (−64%) and σ_b_ (−68%) following wetting were measured with the height gauge. Wetting, therefore, greatly lowers the mechanical resistance of stems. Moreover, the existence of a weak point—i.e., the pulvinus—in mature barley stems is an important feature that must be considered when evaluating the lodging-related characteristics of this crop. These findings improve our understanding of the mechanical properties of barley stems and, thus, our capability to identify genotypes with better lodging resistance.

## 1. Introduction

Lodging is the permanent displacement of the crop stems from their upright position as a consequence of forces exerted by wind, rain, or hail, which can bend over the stems up to near ground level [1,2]. Plants can become plastically (that is, permanently) displaced from the vertical position without fully lodging. The prevalence (percentage of field area affected) and degree (by which the culms lean from the perpendicular) of lodging determine its severity [1]. If severe (i.e., the plants lie flat on the ground), lodging makes a crop difficult to harvest and reduces grain yield and quality [1,3,4].

As the plant is fixed at its base, the wind, usually combined with rain (and the ear weight), produces on the stalk a bending moment (shoot leverage) that is opposed by a bending–resisting moment dependent on the mechanical properties of the stem [1,3]. The degree of stalk bending increases with the bending moment, and up to a certain limit, it is reversible (i.e., elastic), so the plant will resume its upright position as soon as the forces that have induced the bending moment cease to operate [1]. Beyond this limit (namely the maximum bending–resisting moment), bending cannot be reversed, and lodging occurs. The maximum bending–resisting moment is also known as the failure moment or stem bending strength [2,5]. As the stem acts as a lever, the maximum leverage occurs at the stem base. The failure moment of the stem base is, therefore, a key mechanical characteristic of the plant structure [1,3].

Lodging may occur because of failure at either the stem or the root. Stem lodging takes place when the bending moment of the shoot exceeds the failure moment of the stem base and, thus, the stem base buckles at one of the lower internodes of the shoot [2,3]. Buckling occurs suddenly, with negligible prior plastic deformation, and it leads to an almost complete failure of the bending–resisting moment of the culm [2,3]. In root lodging, on the other hand, the leverage of all the shoots belonging to a single plant exceeds the failure moment of the anchorage system (that is, anchorage strength) but not the average stem bending strength; consequently, the root–soil system fails, and the anchorage system is overturned [2,3].

In green herbaceous plants—but not in ripe ones—lodged stalks can partially regain an upright position thanks to the gravitropic response of pulvini, which, by asymmetrically resuming growth, produce an upward bending of the stalk and, thus, return a bent stem to a more vertical position [6]. In the festucoid grasses—which include barley, oat, and wheat—there is a strong gravitropic response in the pulvini, which are conspicuous swellings at the base of the leaf sheath [6] and form a hard knot immediately above the node (Figure 1). Their cell walls have poor secondary modifications, and their sturdiness is chiefly due to turgor pressure [7,8], so they shrink as they dry out (Figure 1D).

In wheat, intact culms are unlikely to break down as long as they are green and turgid. Thus, stem lodging is to be expected chiefly for senescent plants after ripeness [1]. In general, cereal plants become more susceptible to lodging as they approach the end stages of their development, and lodging tends to be more common close to harvest [3]. Dead-ripe barley plants are particularly susceptible to stem lodging caused by storms.

Even though lodging is strongly affected by environmental conditions, genotype effects are also of great importance [1,2]. Cell wall composition, lignin and cellulose have significant positive correlation with lodging resistance of cereal crops and have a largely prevalent genetic determination [9].

In mechanics, ‘stress’ is the internal resistance, or counterforce, of a material to the distorting effects of an external force [5]. The stress (σ) sustained by a material can be equated to the force (*F*) exerted per area of the cross-section perpendicular to the force. The stress can be thus regarded as a pressure/tension [5]. ‘Strain’ is the relative change in the shape of a body that results from the applied force.

Some parameters are of specific interest in studying the mechanics of stem lodging [2] and are described below.

The area moment of inertia, *I*, is the geometrical component of the resistance to bending that depends solely on how the material of the cross-section of the object is distributed relative to the bending axis: the more an object has material located farther from the bending axis the stiffer and stronger it is along that axis. This is why the culms of several grasses are hollow cylinders [7,10]. For these structures, *I* is [5]
*I* = π/4 · (*r*_e_^4^ − *r*_i_^4^),(1)
where *r*_e_ is the outside radius of the stem (that is, half of the diameter), and *r*_i_ is the inside radius of the stem (that is, *r*_e_ minus the width of the stem wall).

For a thin tubular structure, however, Formula (1) can be approximated as [5]
*I* ≈ π · *r*_e_^3^ · (*r*_e_ − *r*_i_).(2)

Although Formula (1) was always used in the calculations of this study, Formula (2) is sometimes used throughout this paper when discussing the relationship observed between some extensive mechanical properties (i.e., those depending on biometric traits) and the biometric traits (namely, *r*_e_ and *r*_e_ − *r*_i_), since the approximate formula is easier to interpret.

Strength is the ability of a body, or material, to resist permanent deformation or even complete collapse. In bending, it corresponds to either a maximum resisting moment of a body or a limit value of stress at which a material ceases to behave according to its function [5]. One of the main mechanical parameters involved in cereal mechanical resistance to lodging is the stem bending strength (*B_S_*), which equals the maximum bending moment the stalk can withstand before structural failure occurs, either by breaking or creeping [2]. Accordingly, *B_S_* is inferred from a bending test as the maximum bending moment before the stem collapses. Theoretically, however, *B_S_* obtains as [1,5]
*B_S_* = σ_b_ · *I*/*r*_e_,(3)
which means that the maximum resisting moment depends on the shape of the cross-section, that is, *I*, and the material strength (σ_b_), which is the maximum stress that a material can withstand before failing, either by breaking or creeping [5,7]. To have *B_S_* in moment units, the product of *I* and σ_b_ must be divided by *r*_e_ (which, for a tubular structure, is the maximum distance from the neutral axis). This also accounts for the fact that the stress is the highest at the periphery of the cross-section; therefore, the maximum bending moment must be scaled down so that the maximum stress to which the material is subjected is not greater than σ_b_ even at the periphery of the section.

Another important mechanical trait is the flexural rigidity, *EI*, which measures the extent to which an object withstands deformation, that is, the force necessary for unit axial strain. It is just given by *E*·*I*, where *E* is the modulus of elasticity, and *I* is the area moment of inertia. *EI* represents the rigidity, or stiffness, of the material, whereas *E* is the slope of the linear part of the stress–strain plot [5].

The three-point bending test is a common method to assess the mechanical properties of the stems [7,11]. Some important considerations have been, however, put forward regarding these tests. Hollow, thin-walled stems (i.e., with a stem wall thickness to stem diameter ratio < 0.15), such as that of barley, collapse due to Brazier buckling [7]. The Brazier effect consists of two consecutive phases: ovalization and buckling [11]. Ovalization of the cross-section takes place at a point of the tubular structure where the stress caused by bending exceeds the rigidity of the stem the most. Hereby, the circular cross-section of the hollow stem becomes elliptical, reducing the effective diameter of the stem and, thus, *I*. Consequently, the maximum bending–resisting moment, *B_S_*, drops too. If the applied force increases further, the ovalization continues up to when the bending moment reaches the dropping effective *B_S_*, and, at such maximum value of the applied force, a kink is suddenly formed on the side of the culm internal to its curvature [11]. Thereafter, the culm bending resistance (in terms of *B_S_*) plummets, and the cross-section locally flattens, causing the buckling of the stem [11]. Unfortunately, in tests, bending a stem with a sharp anvil produces a concentrated pressure that can artificially promote and advance ovalization and kink formation, thereby facilitating buckling [10]. This leads to an underestimation of the actual mechanical resistance of the stem. In bending tests, it is therefore necessary to avoid fostering the Brazier effect. To this aim, it has been recommended that these tests should be conducted by loading the force at the node rather than at the internode, since the former is filled (because of the nodal septum) and, thus, it does not ovalize [10].

The present work aims to identify variables that affect stem mechanical properties, and, thus, the evaluation of genetic differences in lodging-related traits. In this way, we sought to understand why, commonly, stem mechanical properties have only a partial capability to predict lodging resistance, which suggests that some relevant variables are missing in bending tests. The identification of these missing factors can improve our understanding of lodging resistance. Apart from an initial experiment in 2021, in which green stems were tested, fully mature and dry culms were used in this study because late-season stem lodging typically occurs near harvest time when the plants have senesced and dried down [1,3,12]. It was hypothesized that, if it is not uniform, moisture content could have a large confounding effect on the measurement of strength in ripe stems. Thus, the stems were left to dry, equilibrating with ambient moisture, for a long time before testing. For comparison, the dried stems were also fully wetted overnight. Once it was clarified that stem moisture content indeed has a large effect on the mechanical strength of the culms, it was further evaluated how loading at the internode rather than at the node affects the results. This was performed by manually (i.e., with a fingertip) loading the stem segments in the three-point bending test. Even the ‘loading point’ factor showed a remarkable effect, the implications of which are discussed in this paper.

## 2. Results

Experiments were performed with green plants in 2021 and with ripe plants grown in 2022 and 2023.

### 2.1. Bending in Green Barley Plants (2021)

In 2021, the measurement of stem failure (at the early dough stage) was performed with a height gauge immediately after sampling each stem segment. Results are shown in Table 1. The moisture content of the basal, mid, and neck stem portions was, on average, 68.0%, 71.1%, and 73.7%, respectively.

The upward decrease in *I* along the stem was largely due to the reduction in the width of the culm wall. From the base to the neck node, *B_S_* decreased because of the reduction in both *I* and σ_b_.

In green plants, failure of stem segments (by applying the stress force at the node) occurs neither at the node nor at the pulvinus, but below the node (Figure 2).

### 2.2. Bending in Ripe Barley Plants

In 2022 and 2023, experiments were set up to test bending of ripe stems.

#### 2.2.1. Comparison of Dry and Wet Ripe Stems with Height Gauge (2022)

In 2022, the measurements of stem failure were made with a height gauge in the late Autumn. The results are shown in Table 2. Ripe stems were used for this, as well as all the subsequent experiments. To equalize their moisture content (which can vary because of both differences in the growth cycle between genotypes and random differences among plants), basal segments were cut off the stems and stored at room temperature in open vessels to equilibrate with ambient moisture for a few months. As harvest time is a very busy period for breeders, this delay also allows testing at a less frantic time. Additionally, to evaluate the effect of moisture content on the stem mechanical properties, part of the stored stem segments was fully wetted overnight. The moisture content for dry and wet stem segments was, on average, 9.0% (range across genotypes: 8.6–9.4%) and 69.4% (range across genotypes: 67.9–72.7%), respectively.

The stem bending strength (*B_S_*) dropped −64% when stems were wetted (Table 2). This was essentially due to a corresponding (−68%) plunge of their material strength (σ_b_). *E* decreased by a lower extent, −41%, following wetting. Figure 3 shows that, though having a large effect on the absolute values, testing either dry or wet stems had no effect on the ranking of genotypes. It is obviously important, however, to compare stems that have approximately the same moisture content; otherwise, stem moisture acts as a strong confounding factor. Both leaving ripe stems to dry at room temperature for some months or subsequently wetting them overnight appear to be reasonable solutions to equalize their moisture content.

Correlation analysis was performed separately for dry and wet stems, as this factor strongly affects the mechanical behavior of the stems (Table 2). Table 3 shows the correlation for dry stems. A graphical representation and the corresponding analysis for wet stems are given in the Appendix A.

There was a positive correlation between the outer stem diameter, *D*, and wall thickness, *r*_e_ − *r*_i_, as well as between these two traits and *I*, which, indeed, is calculated from the former biometric traits. Significant positive linear correlation was observed between most mechanical parameters. A negative correlation was found between the modulus of elasticity (*E*) and biometric parameters, particularly *D*, as well as between *E* and *B_S_*. Negative correlations between *D* or *r*_e_ − *r*_i_ and *E* were also found by Skubisz [13] in wheat. A negative relationship between *E* and *D* suggests a physiological compensation effect, that is, thinner culms tend to have higher material rigidity. No significant correlation was found between *E* and *EI* because of the negative relationship between *E* and *I*. Material strength (σ_b_) was poorly, or not significantly, correlated with biometric parameters, as well as with *E*. Nevertheless, it slightly positively correlated with *EI*. In wet stems, correlations were largely similar, but higher and significant positive correlations were observed between σ_b_ and biometric parameters (Appendix A). It might be worthwhile to notice that the relationship between *I* and *D*, even though already very high, is underestimated by the Pearson correlation coefficient because it is a cubic relationship (see Formula (2)), rather than a linear one (Appendix A). Overall, the most noticeable relationship is the negative one between *E* and biometric parameters, and, therefore, between *E* and *B_S_*: their opposite variation among genotypes is evident in Table 2 and suggests that high strength is associated with low material stiffness. This could be seen as barley stems requiring both strength and flexibility so that they do not break down easily.

Bending strength also depends on σ_b_, and, as σ_b_ and *E* are uncorrelated (Table 3), diversity in the material properties of the genotypes can be further evidenced by using the ratio between σ_b_ and *E* (Figure 4). This ratio, indeed, represents the strength of the material over its stiffness, that is, the capability of a stalk material to withstand bending without failing rather than its tendency to oppose a rigid resistance to the wind force. Indeed, flexibility enables the stem to shed wind loads by bending downwind, thereby reducing wind drag [7].

In the present study, the σ_b_ to *E* ratio showed the same genotype ranking as *EI* and *B_S_* (Table 2 and Figure 3), pointing out the poor overall mechanical properties of cultivar Istos (Figure 4). Thus, this ratio can be used to evidence genotype diversity in straw material properties.

A noteworthy general issue with data of culm mechanical properties was the high variation observed among stem replicates. This was already noted with green plants in the previous year, but as our aim was to specifically investigate ripe plants, we investigated this in more detail using data from 2022 and 2023 experiments. An insight on this topic is provided in Appendix B.

#### 2.2.2. Comparison of Dry Ripe Stems with Height Gauge (2023)

In 2023, the experiment was repeated with four blocks. The first block was used to replicate the testing procedure on dry culms exactly as in the previous experiment (2022), that is, using the height gauge, whereas the other blocks were employed to evaluate whether the procedure could be shortened by making it entirely manual. Tests were performed in early 2024 after leaving the stem segments to dry at room temperature for several months.

Results of the measurements made with the height gauge on dry ripe stems are presented in Table 4. The average moisture content of the stem segments was 6.6%.

From Table 4, it can be envisaged that, though the values of most parameters are decidedly lower than in the previous year (suggesting a relevant year effect), the ranking of genotypes was consistent, which is a requisite for any attempt at identifying stable genetic effects. Intensive mechanical properties (namely, *E* and σ_b_) were not significantly different among genotypes, whereas significant differences were found for all the extensive properties: Ketos and Tibet-A4 consistently had the highest values, Fior6814 was second, and Istos was last. These data confirm that the main genetic differences in the strength (in terms of *B_S_*) of the dry stem are essentially due to larger and thicker stems. The modulus of elasticity (*E*) and flexural rigidity (*EI*) did not add relevant information to discriminate genotypes over and above that provided by *I* and *B_S_*.

#### 2.2.3. Comparison of Dry and Wet Ripe Stems Loaded at the Node or Internode with the Manual Method (2023)

It was then evaluated whether assessing the mechanical features of stems manually, rather than by means of a height gauge, could be reliable enough to compare different genotypes. As a height gauge is needed to measure *E* and *EI*, but these parameters were not absolutely necessary to our purpose, a manual assessment was deemed to be a valuable approach, because it only requires a scale as instrumental support for the bending test, and it can save a lot of time.

In this experiment, therefore, the assessment of stem mechanical features was performed by manually applying the stress force with a finger, instead of the height gauge, in the three-point bending test. Since pressing with the fingertip provides a soft, ample surface, it was assumed that this did not foster the Brazier effect over the natural mechanical failure, and, therefore, results obtained by targeting either the node or the internode as loading point were considered comparable.

Manually applying the stress force with a finger was performed carefully, trying to apply a pushing force increasing at an even rate, so to lead to stem failure within the same time (1 min target; about 0.5–1.5 min effective range) as by using the gauge. The time was easily equalized with the previous method, empirically managing it with the same criterion. Thus, in this experiment, the effect of using dry or wet stems, as well as the effect of targeting either the node or the internode as a loading point, were evaluated.

As observed in the previous experiment with dry/wet stem segments (namely, the 2022 experiment; Table 2), the moisture content caused a large drop in the material strength (σ_b_) of the stems (Figure 5). Stem moisture is, thus, confirmed as a key factor when the mechanical properties of ripe culms are being assessed.

The results of the test with manual assessment are presented in Table 5. In this experiment, the moisture content for dry and wet stems was, on average, 6.6% (range across genotypes: 6.0–7.5%) and 61.2% (range across genotypes: 56.9–65.3%), respectively.

Table 5 displays the overall effects of using dry vs wet stem segments, the effect of applying the bending force at either the node or internode for dry and wet stem segments separately, and the genotypic values at the four combinations of testing conditions.

The stem bending strength (*B_S_*) dropped −62% when stems were wetted. This was essentially due to an equivalent (−62%) plunge of their material strength (σ_b_). These results are remarkably similar to those obtained in the previous year with the height gauge.

When the bending stress was applied at the node, a lower *B_S_* was observed in both dry (−31%) and wet (−41%) stems with respect to loading at the internode. Apparently, this was in part due to a reduction in the observed σ_b_ (−11% and −25%, respectively) and to the lower diameter at the node, which caused a drop of *I*: −33% in dry stems and −65% in wet ones. Unfortunately, this interpretation is not correct: although the bending stress force was applied at the node when it was targeted as a loading point, the stem failure—in this case—occurred at the pulvinus (Figure 6). Calculations of σ_b_ for the node should apply, therefore, to the pulvinus, but they are flawed, because the pulvinus has a different structure and diameter with respect to the node, and, thus, its σ_b_ can be, and probably is, different from the calculated one. If it were possible, taking biometric measures at the pulvinus would have been the right way to assess its properties (once one knows that it consistently is the failing point), but the pulvinus structure is complex, and it varies considerably in the space of less than 1 mm. For a precise evaluation of the properties of the pulvinus, it would be necessary to remove the sheath, because the exact failing point is not visible otherwise; however, the thickened sheath is a main part of the pulvinus, and it is structurally important for stem stiffness and bending strength [14]. It cannot, hence, be removed without profoundly altering the mechanical properties of the pulvinus. Nevertheless, the largely lower *B_S_* observed when the node rather than the internode was targeted as a loading point, combined with a corresponding systematic failure at the pulvinus in the former case, is a stark indication that the pulvinus is the weak point for bending resistance in ripe barley stalks.

Thus, assessing the mechanical properties of a stem at the node/pulvinus complex is not the right approach when one wants to obtain a reliable measure of σ_b_, but it is the approach to take if one wants to ascertain the role of the pulvinus in the bending strength of the stem, that is, if *B_S_* values are the chief focus.

Appendix C shows several noticeable properties of these data emerging during their elaboration. One important aspect is that many parameters—particularly *D*, *I*, and *B_S_*—display a lognormal error distribution, which makes it more proper to compare these data on a logarithm scale. The means of the natural logarithms of *B_S_* are, therefore, displayed in Figure 7 for the best comparison of the effects of testing conditions.

Figure 7 shows that, though having a large effect on the absolute values, the different testing conditions remarkably had no effect on the ranking of genotypes. They can, therefore, all be used as alternative conditions to compare barley genotypes. However, it is obviously necessary to compare genotypes using the same loading point and stems with the same moisture content. The easiest approach is to let the stem segments dry out to equilibrium with the room atmosphere before testing.

Figure 8 shows that stem thickness (*r*_e_ − *r*_i_) is roughly linearly related to the stem diameter (*D*), that is, large stems are also thicker. However, this relationship depends on whether the loading force is applied on the node or the internode, because stem thickness is always greater close to the node. As the stem is filled at the node, stem diameter and stem wall thickness were measured 2–3 mm below the node. A positive, linear relationship between the diameter and wall width of basal internodes was also found in wheat [15].

As seen, *B_S_* depends on the area moment of inertia (*I*), as well as on the material strength (σ_b_), and the latter is highly sensitive to the stem moisture content. Since *I* is roughly proportional to the third power of *D* (Formula (2)), *B_S_* shows a quadratic relationship to *D* (as it depends on *I*/*r*_e_; see Formula (3)) that is basically dependent on the stem moisture (Figure 9A). The effect of the loading point, either the node or internode, was significant (Table 5), but it did not affect the relationship between *B_S_* and *D* (Figure 9B).

Figure 9B shows that the node, even if it is associated with a thicker wall in the terminal part of the internode immediately below it (Figure 8), is also associated with a lower *B_S_* in the bending test; but, as seen, this is due to the failure of the pulvinus adjacent to the node.

### 2.3. Stress–Strain Diagrams

Although, in a three-point bending test, the actual stress and strain ought to be calculated from the applied force, the observed displacement, and the biometric traits [12], it is common practice to refer to the measurements, and plots, of the applied force and the observed deflection as those of stress and strain [5]. Thus, here, stress–strain diagrams (actually, force–deflection plots) are intended as a representation of the force necessary to produce a corresponding deflection during a three-point bending test.

Throughout this study, mechanical parameters were assessed trying to maintain a relatively similar time, which roughly corresponds to a constant displacement increase, across tests. For stress–strain diagrams, however, a more well-defined, fixed increment of the bending displacement was adopted because the software collects data on the applied force at constant time intervals. Thus, a constant increase in displacement is required to produce these diagrams, which can be obtained only using the height gauge.

Figure 10 illustrates exemplificative stress–strain diagrams for dry and wet stems. The shape of the diagrams showed large random variation, both in relation to differences in the modulus of elasticity (that is, the slope of the initial linear increase), the maximum strength (of which *B_S_* is a linear transformation so that they have the same high coefficient of variation, which is provided in Appendix B), as well as to the way failure took place (sometimes abrupt failure occurred in two steps). However, some features were consistent: (1) A large initial portion of the curve displays a linear relationship between force and deflection, thereby confirming that the stem stress response is largely proportional to strain, and, thus, the initial deformation is broadly elastic (as a linear response indicates that the properties of the material do not change). (2) The range of the linear response (up to the proportional limit) is about twice in dry stems with respect to wet stems, which suggests the latter are more prone to plastic deformation. Besides, (3) in dry stems, breaking (i.e., a sudden collapse in the stress response to the increasing strain) is typically observed shortly after the stress has reached its maximum, sometimes even without a prior gradual decline. (4) Breaking is, instead, not observed in wet stems. The descent to lower stress levels at high strain is smooth and takes place as a gradual and prolonged creeping failure when the stems are wet.

As the most common cause of lodging is a storm, it causes ripe plants to soak up water and, thus, they have a higher probability of permanent displacement of the stem from the vertical than green plants, since their deformation is plastic for a much larger portion of their response curve than occurs in green plants. Ripe plants have, therefore, greater chances of incurring stem lodging of low degree (that is, with a moderate displacement from their original vertical position). In addition, they also have a greater probability of full lodging, since wet, ripe culms have a lower *B_S_*, and moderate lodging prompts further lodging [1,3].

## 3. Discussion

Our study highlights the huge negative effect of high moisture content on the mechanical resistance of barley stems once the whole plant has dried out and the crop attains full ripeness (Figure 3, Figure 4, Figure 5 and Figure 7). Wetting of ripe stems, such as due to heavy rainfall, is, thus, expected to cause a plunge in the strength of the straw. Although demonstrating that straw becomes less stiff and loses strength when wet would seem to state the obvious, the quantification of this phenomenon is nevertheless useful to elucidate a component of the complex set of factors that cause lodging.

These results emphasize the great importance of equalizing stem moisture as much as possible before testing stem mechanical properties. Leaving the stem samples to dry out in open containers for some months appears to be a convenient way to achieve this objective. If this step is not performed, disformed moisture content about individual stem replicates could make the data unsuitable for reaching significant results. In a preliminary experiment, we observed, at physiological maturity, overall variability in the moisture content of basal portions of main stems in the range of 20–70%, with both differences among genotype averages and wide random variation within genotypes. In a dead-ripe barley standing, stem moisture content is lower, but large differences due to weather, soil conditions, and random variation are expected.

Our results are in accordance with the study by Annoussamy et al. [16], who found that (as an average over four internodes at different positions along the stem) wheat σ_b_ decreased by −58% and *E* by −37% when dry culms are wetted. Those authors also found that this reduction was already completed (i.e., saturated) at about 30% moisture content and remained stable through higher moisture content. Smaller but significant decreases in σ_b_ and *E* caused by a 5–10% increase in moisture content were found by Tavakoli et al. [17] for barley stems collected at harvest time. In wood, as in straw, absorption of moisture up to fiber saturation greatly lowers *E*, σ_b_, and the proportional limit [5]. The fiber saturation point is the point above which any change in moisture content has little effect on mechanical properties but below which strength and stiffness increase substantially with decreasing moisture content [7].

It is worth noting that, while the largest effect of differences in moisture content is on σ_b_, and the main differences among genotypes relate, instead, to culm diameter and thickness, the old cultivar Istos also diverges from the other genotypes as regards σ_b_ (Table 2, Table 4 and Table 5) and the σ_b_ to *E* ratio (Figure 4). Thus, even though the biometrical traits have a dominating effect on differentiating stem mechanical properties among genotypes, noticeable genetic variability in stem material properties exists too.

For the most common purposes, genetic differences in the resistance to stem lodging are mainly determined by just two biometric parameters: stem diameter and plant height [2]. As seen, the stem radius (*r*_e_) has a squared positive effect on *B_S_*, since the bending strength of thin-walled tubes is proportional to *r*_e_^2^·(*r*_e_ − *r*_i_), given it depends on *I*/*r*_e_ (Formula (3)) and *I* is a cubic function of *r*_e_ (see Formula (2)). Plant height, on the other hand, has a linear to quadratic effect on the basal bending moment caused by wind, depending on whether the wind force vector orthogonal to the culm is concentrated at the top (i.e., the ear) or is diffuse along the entire stem span, as can be seen by considering the plant as a vertical cantilever beam [5,8].

Historically, reducing crop height has been the main avenue to reduce the lodging risk in cereals [2]. In modern small-grain cereal cultivars (which have been selected for an average height spanning 0.7–1 m), however, the risk of stem lodging is mostly influenced by stem diameter [2]. The relevance of stem diameter is strengthened by its positive correlation with culm wall width in both wheat [15] and barley (Table 3 and Figure 8). In barley, differences in culm wall thickness have also frequently been correlated with varietal differences in lodging resistance [2]. In a recent study with fully mature barley, lodging showed significant negative correlations with outer diameter and section modulus (*I*/*r*_e_) but no correlation with plant height [18].

It is also worth noticing that, though the absolute values of the mechanical parameters—particularly *B_S_*—showed variation among the experiments, the ranking of genotypes was consistent across both testing conditions (Table 5 and Figure 3 and Figure 7) and experiments (compare Table 2, Table 4 and Table 5). This indicates that, for the assessment of genetic differences, researchers can choose the best testing condition for their purposes: either a manual method, which does not require the use of a height gauge and is therefore faster, or a more precise assessment with the height gauge, which allows the additional evaluation of *E* and *EI*. The manual assessment of the stem mechanical properties is cheap in terms of required devices, faster and can be easily introduced in any breeding program. The height gauge can be used when a more precise characterization is required, such as if divergent genotypes are found after a preliminary assessment. Moreover, notwithstanding *B_S_* is the main mechanical parameter for stem lodging, it usually chiefly depends on the diameter of the culm and its wall thickness. Thus, it is sensible to measure *B_S_* and σ_b_ only when searching for genotypes with an unusual material strength.

It has been remarked that the internodal-loaded three-point bending test can produce erroneous bending strength measurements because, in hollow stems, the Brazier effect may be artefactually fostered: as the anvil presses the stem on a small surface, the stem cross-section becomes more oval sooner than if a diffuse force were applied, reducing its ability to resist bending [7,10,11]. It was, therefore, recommended that the anvil be loaded at the stronger and denser nodal tissues and not in the middle of an internode. Nonetheless, by manually applying the force with the fingertip, we found higher bending resistance in the middle of the basal internode rather than at a corresponding basal node (Table 5). We assume this supports that the Brazier effect (which could otherwise cause an underestimation of *B_S_* when loading at the internode) was not artefactually promoted with this method thanks to the wide and soft surface with which the loading force was applied to stem segments. A largely lower value of *B_S_* when loading at the node with respect to the internode further supports that it is advisable to load the force at the node in a bending test: as the mechanical resistance of the pulvinus limits the overall resistance of a ripe culm, measuring at the weak point provides a more realistic evaluation of stem resistance.

When stems are tested for their mechanical properties with nodal loading, a systematic failure at a site different from that of maximum leverage (which, in a three-point bending test, is at the mid-span) indicates that such site is weaker than the part of the stem that was subjected to the maximum leverage. This is the case for the pulvinus in ripe barley (Figure 6B). Figure 2 shows that, instead, when the node of green stems is targeted as a loading point, bending occurs neither at the node nor at the pulvinus but at the internode just below the node. This means that, in green plants, the node–pulvinus complex is more resistant to bending than the internode. Hence, as regards stem bending, the pulvinus is the weak point in ripe plants but not in green ones.

Although we have not performed a rigorous assessment of the changes in the mechanical properties of barley culms throughout the plant cycle, our preliminary results to this regard confirm that higher values of both *E* and σ_b_—which, in turn, lead to higher values of *EI* and *B_S_*—are observed in dry ripe stems (Table 2, Table 4 and Table 5) with respect to green ones (Table 1). In accordance, *E* increases in wheat during ripening [13,19]. This notwithstanding, in our experience, barley is no less susceptible to stem lodging as a ripe plant than as a green one. Based on our present results, the reason for this inconsistency lies in the dramatic loss of mechanical resistance of the pulvinus when it loses turgor and dries out. In addition, when the crop is ripe, wetting of the plant by rain severely further lowers the mechanical strength of the culm. Even though the cereal stem obviously is a heterogeneous structure, it would seem that a difference in the mechanical response of the internode and the node complex had not been reported before for ripe barley.

In green plants, turgor pressure has a substantial positive effect on measured mechanical parameters such as *E*, *EI*, σ_b_ and *B_S_* [7]. Herbaceous plants, indeed, support their bodies using the turgor pressure caused by water internal to the membranes of their living cells [8]. Thus, a drop in turgor pressure when plants suffer water stress leads to wilting, abating the values of the above-mentioned mechanical parameters [7]. However, in ripe plants—which no longer have a turgor pressure—an increase in moisture content results in a decrease in *E*, σ_b_, and *B_S_*, as found in both wheat [16] and alfalfa [20].

In a vertical cantilever beam fixed at its basis (like a barley plant anchored in the soil), failure occurs at the fixed end, where the bending moment is greatest [5,8]. However, barley—differing in this from wheat—often shows stem failure at the middle internodes. Such phenomenon is known as brackling [3]. If the *I* and, more importantly, the *B_S_* of a cantilever beam decrease from the fixed end to the free end (Figure 11), the failure point can change significantly with respect to a beam with uniform *I* [5]. Berry et al. [21], therefore, proposed that a steeper reduction in *I* and *B_S_* up the stem, together with greater flexibility of the barley stems, which reduces the bending moment that is transmitted down to the base of the stem [3], can be the cause of brackling in barley.

Brackling can be of interest in the present study because, whereas the nodes and internodes below the soil surface are difficult to examine, brackling makes it easier to assess the exact point where failure has occurred. This is important because Robertson et al. [10] recommended that the types and location of failure observed in the bending test should be similar to those found in the natural loading environment. Figure 12 shows that, when brackled plants are examined, field observations are consistent with the type of failure expected on the basis of our experiments: on the one hand, when the plant is green, stem buckling happens along the internode, at least half a centimeter below the node/pulvinus complex (Figure 12A–C). Although failure can happen even lower along the internode, the presence of the leaf sheath, which tightly envelops the stem and thus strengthens it, makes this occurrence less probable. Anecdotal observations also suggest that, if the plant is partially dry, so that the leaf and sheath laminas but not the pulvinus have dried out, stem failure can sometimes happen just above the pulvinus. Failure, however, practically never takes place at the pulvinus until it stays green. On the other hand, when the plant is dead-ripe, stem failure often—though not always—occurs at the pulvinus. The pulvinus fails by buckling in dry stems (Figure 6) but by creeping in wet stems (as occurs due to wind and persistent rain), which become inflexible again when they re-dry (Figure 12D–F). Relevant statistical data on where and how exactly stem failure occurs when barley lodges in the field are not available, unfortunately. Nonetheless, these preliminary observations are encouraging regarding the capability of our results to be representative of the features of stem lodging in the field.

It is interesting to compare the presently observed values of stem mechanical properties with those found in previous studies. For green barley plants between milk and dough grain stages, Berry et al. [21] found *EI* values in the 13.2–24.6 mN·m^2^ range at the middle section of the stem. Except for cultivar Istos, we found values lying within this interval at the same stem position (Table 1). The same authors found *B_S_* values in the ranges of 52–75, 27–42, and 18–22 N·mm for the bottom, middle, and top portions of the stem, respectively. In this case, our values (Table 1) are either lower (for cultivar Istos) or higher (for the other three genotypes) than these intervals at each position. The cultivars used by Berry et al. [21] were, evidently, more uniform than our genotypes. For barley stems at a stage and moisture content similar to those characterizing ripe plants in our study, Leblicq et al. [11] found a value of 3.58 GPa for *E*, which is well comparable with the values found in our study (Table 2 and Table 4).

As mentioned above, barley has noticeably lower flexural rigidity and stem failure moment than wheat [21]. To illustrate the differences between the two crops, for wheat, it has been recommended that plants should have [2] a height < 1 m and as close as possible to 0.7 m (the minimum height compatible with high yields), a stem diameter of 4.94 mm, a wall width of 0.65 mm for the bottom internode, and a material strength of 30 MPa. Clearly, both stem diameter and wall width are lower for barley (Table 1). In other words, *I* is limiting for barley. As a high value for *EI* may be due to both *E* and *I*, improving the latter would seem the best way to increase lodging resistance in barley. In accordance, varietal variation in flexural rigidity is mainly due to high *I* in barley, whereas it originates primarily from a high *E* in wheat [1].

Regarding the material strength, green plants (Table 1) have suboptimal values for this trait too, but dry ripe ones have higher values (Table 4 and Table 5). Nevertheless, even dead-ripe plants have a too low material strength if wet, particularly as measured at the node complex. Regarding *B_S_*, it was reported to vary from 122 to 175 N·mm among modern wheat cultivars [2]. Although green plants of some barley genotypes reach the low end of this range (Table 1), wet ripe stems greatly miss this goal (Table 4 and Table 5).

Tabaracci et al. [12] found a strong linear relationship between stem flexural stiffness and bending strength (r = 0.96) across culms of the same plot of spring wheat (indicating a constant ratio between *B_S_* and *EI*). However, we found a significant but weaker correlation (r = 0.67) between *B_S_* and *EI* in dry barley stems of different cultivars (Table 3). A certain degree of correlation between these two parameters is expected because they both depend on *I*, but, as previously remarked, σ_b_ and *E*—which *B_S_* and *EI* also depend upon, respectively—are not correlated across barley genotypes (Table 3). Nevertheless, the genotypic averages of *B_S_* and *EI* show a positive linear relationship (r = 0.98 in 2022 for dry stems, as well as in 2021 for basal green stems; correlations are for genotype averages from Table 2 and Table 1, respectively). On the one hand, a strong linear relationship between *B_S_* and *EI* across genotypes suggests that the two parameters might be redundant for genetic discrimination, at least under the studied experimental conditions. On the other hand, our results indicate that the ratio between σ_b_ and *E* has a noteworthy genetic determination (Figure 4), at least within a given environment. Thus, as mentioned above, measuring *E* can be useful, but chiefly after relevant genetic variability for σ_b_ has been ascertained.

In green and dry cereal plants, buckling occurs suddenly, with little plastic deformation [2]. In accordance, the modulus of elasticity of wheat and barley remains reasonably constant, even quite close to structural failure, which indicates that the limit of proportionality between the applied stress and corresponding strain is seldom exceeded before failure [2]. A tiny interval for plastic deformation in dry stems was confirmed in our study (by applying the force at the node), but such an interval was much larger when the stems were wet (Figure 10). We also observed that breaking does not occur in wet stems; rather, they display a gradual creeping failure at the pulvinus (Figure 10B and Figure 12D–F). Oddly, this phenomenon went unnoticed in past studies.

In general, a limitation of studying the mechanical properties of cereal stems is that good stem mechanical resistance does not prevent root lodging. Thus, mechanical parameters like *E*, *EI*, σ_b_, and *B_S_* usually display only a mild correlation with lodging resistance [1]. In addition, an odd issue is that even an obvious mechanical trait like *B_S_* does not explain differences in lodging resistance of the landrace Tibet-A4, which has a high *B_S_* but always lodges at maturity, even in the absence of strong winds. Although this mismatch deserves further investigation, having a good method for evaluating mechanical properties clearly associated with lodging resistance and considering variables like moisture content and the effect of pulvinus are important advantages for a more complete evaluation of lodging resistance.

Some methodological issues are of concern when three-point bending tests are performed, and they are briefly considered here in relation to our testing procedure. Robertson et al. [10] recommended that (i) stem span lengths longer than 20 times the diameter of the stem are employed in the three-point bending test; (ii) loading anvil and supports with rounded edges rather than sharp corners are used; (iii) both nodal and internodal tissues be flexed in each test; and (iv) the rate of deflection should minimize viscoelastic effects (that is, both stress relaxation at very slow speeds and strain hardening at very fast speeds should be avoided). As regards the last point, Shah et al. [7] stated that, as a rule of thumb, in a bending test, failure time should be ~60–90 s. In our experiments, we achieved failure of stem segments in about 1 min when the downward displacement was around 6 mm (Figure 10).

Shah et al. [7] also recommended that any analysis of stem mechanical properties, as well as the resulting statistical analysis of the data, should be based on several repeats (at least 10) over a number of experimental replicas (at least three). Piñera-Chavez et al. [22] found that a minimum of seven plants per plot (with each treatment plot replicated three times) is required to identify genetic differences in the mechanical properties of lodging-associated traits of the main shoot. In our study, 5–6 basal stem segments for each of the three block replicates were sufficient to reveal significant differences among genotypes within each combination of wet/dry and node/internode testing conditions (Table 5; Figure 7). A problem affecting the analysis of mechanical properties and their effects on lodging is, indeed, that there is large variability among plants, since these parameters are strongly affected by environmental and random factors, whereas varietal differences within the same species and maturity class often are rather slight [1]. This is immediately evident by looking at the CVs of the mechanical parameters (see Appendix B). Correspondingly, Tabaracci et al. [12] found that, within the same plot of spring wheat, *EI* and *B_S_* (in stems with leaf sheath retained) showed wide ranges, namely 48.2–248.0 mN·m^2^ and 135.2–641.0 N·mm, respectively. An accurate statistical analysis is therefore required to manage the high uncertainty of estimates associated with the strong experimental variability typically observed for measures of stem mechanical properties.

Four barley genotypes with largely different features of stem traits were used in this study. As these barley genotypes are very different, phenotypically and genetically, we deem that our findings reveal fairly general features of barley. To determine the extent to which this is representative of the characteristics of barley varieties worldwide, a more extensive study on many genotypes is, of course, needed.

## 4. Materials and Methods

### 4.1. Barley Materials

Four barley genotypes were grown in a greenhouse at CREA—Research Centre for Genomics and Bioinformatics in Fiorenzuola d’Arda (Piacenza, Northern Italy)—in a randomized design in 2020–2021 and in a randomized complete block design with three experimental blocks in the growing season 2021–2022 and four in 2022–2023.

Descriptors of these genotypes are provided in Table 6. Data on plant height are from the 2021 experiment. Lodging resistance scores of ripe plants, measured on a 1–9 scale (where 9 is very lodging-resistant [23]), were estimated from field observations through several years with multiple occurrences of lodging. Tibet-A4 was assigned a score of 1 because it always lodges at maturity, even in the absence of severe wind episodes; green plants of this genotype are, however, much less susceptible to lodging.

The seed was dressed with Redigo Pro (Bayer Crop Science, Monheim am Rhein, Germany). The four genotypes were grown in pots (square, 15 × 15 cm, 20 cm high) containing about 3.5 L of growth substrate (Special TNA 2; Vigorplant Italia srl, Fombio, Italy). Five pots per genotype were present in each block replicate, with 3–4 plants per pot. Sowing was performed in the last week of November. The temperature in the greenhouse never fell below 0 °C. Plastic stem supports were used to avoid bending of plant stems when handling the pots. Water was added 3–5 times a week by filling the pot saucer. When the plants were fully ripe in the following summer, segments about 13 cm long were sampled from the stem. Apart from the 2020–2021 experiment, a single segment was collected from the stem base approximately 1–4 weeks after harvest maturity. As there are huge differences between tillers and the main shoot, and among tillers [1], we aimed at collecting only the main culm of each plant, but when it was not immediately obvious which stem was the main one, the two strongest and tallest stems were collected and put in 1 L glass jars without cap. The jars with the stem segments were covered with a layer of Miracloth (Calbiochem, San Diego, CA, USA), held in place with a rubber band, and stored at room temperature until testing. Fully mature and senesced barley stems were therefore used in all experiments except for the 2021 experiment, wherein stems from green plants were used.

### 4.2. Stem Segment Measurements

A three-point bending test was used to assess stem bending properties. Stem segments were cleared of the leaf sheaths, except the part above the node, and they were placed on a support made by carving two rounded slots at opposite positions in the top edge of a round plastic box, 90 mm in diameter. The support was put on a precision balance (LP-2102i, VWR International, Milano, Italy; weighting capacity 2200 g) to measure the applied stress force. In initial experiments, an analogic double-column height gauge (MicroMet; Rupac srl, Milano, Italy) was used to apply the stress force while measuring vertical movements during the three-point bending test. An ad hoc-made L-shaped brass arm with a wedge-shaped tip (covered with a rubber layer to reduce shear stress) was locked by means of the stylus clamp to the slide of the height gauge to act as a loading anvil for the application of the stress force. Before each test, the edge of the gauge wedge was carefully approached to the stem. Then, the test was started by gradually lowering the slide bearing the wedge at 0.1 mm intervals and recording (from the balance) five force measures in sequence for the determination of *E*. The wedge was then gradually lowered to read the maximum value of the loading force. As recommended [7], the test was conducted to last about 1–1.5 min, on average, though the actual time could be as low as 0.5 min for very weak culms.

Biometric stem measures were made with a caliper (CoolantProof IP67 ABS digimatic caliper; Mitutoyo Italiana srl, Milano, Italy) after the bending test. To avoid that deformations caused by the stress test could affect the measurements, measures were taken about 2 mm below the loading point. The mean of four measures of the stem diameter (*D*)—taken by rotating the stem about 45° after each measurement—was used for calculations. The stem was then transversally cut at the same position, and the median of five measures of stem wall thickness (*r*_e_ − *r*_i_) was used for calculations.

When wet stems had to be tested, stem segments were randomly taken from a jar and soaked overnight in 50 mL cylinders containing 10% ethanol in water. The stem pieces were kept fully submerged by means of gliding plungers with handles, and they were individually removed from the bath and briefly blotted with a paper towel immediately prior to testing.

Moisture content was assessed by weight difference after drying stem segments at 105 °C for 24 h. Values of moisture content are given on a wet basis.

To develop stress–strain diagrams, the scale was connected to a laptop equipped with i-Weight software (version 2.0; BEL Engineering srl, Monza, Italy), which allows time kinetics to be performed, acquiring weight values from the scale at predetermined time intervals. A constant downward displacement increase was obtained by manually applying (to the node) an increment of 0.1 mm per second with the height gauge. For both dry and wet stem segments, three to four diagrams were acquired for each genotype.

### 4.3. Experiments

In 2020–2021, an initial experiment (shortly, the 2021 experiment) was set up with green plants, and stem portions were sampled at the early dough stage from plants sown in 2020. As it is recommended to target the node as a loading point when a mechanical pointer is used in the three-point bending test to avoid fostering the Brazier effect [10], and the node was centered on the bending support for testing (as this standardizes the calculations), it was necessary to choose each 13 cm length so that it had a node roughly in the middle when it was collected from the stem. Three lengths were cut from each stem: a basal segment, centered around the lowest node for which a relatively straight-up segment could be obtained; a middle segment, centered around a node about in the middle of the stem (from the base to the neck node); and a neck segment, centered around the neck node. Stem segments were collected from the last decade of April to the first decade of May 2021, and bending properties were immediately assessed after each stem was sampled.

In the 2021–2022 experiment (shortly, the 2022 experiment), only basal segments were collected, with the same procedure as in the previous experiment: as the 13 cm length had to be roughly centered on a node when cutting it from the stem base, when necessary, the actual position where the basal segment was cut from the stem was shifted upward along the stem until a node fell in the center of the segment. Stem segments were collected on 11 July 2022, and bending properties were assessed on 8–18 November 2022. About ten basal stem segments were assessed for each genotype and dry/wet condition.

In 2023, two experiments (shortly, the 2023 experiments) were performed with the plants sown in 2022: one on the first experimental block and the other on the three remaining blocks. Ten stem basal segments were assessed for each genotype. The first block was used to collect basal stem segments and assess their properties with the same method used in the previous year. For the other three blocks, stem segments were, instead, collected and tested with a different procedure: the basal stem segments were always cut at the actual lowest position along the stem, independently of the relative position of nodes. Sampling of basal stem segments was performed on 21 June 2023, and the stem segments were stored as above until testing (between 26 March and 16 May 2024). Ten to twelve stem basal segments were tested for each combination of genotype and dry/wet testing condition. Since the stem segments could have either a node or an internode at their midpoint, the most suitable structure between these two was centered on the support each time, and the stress force was then applied to it. Hence, the node/internode (N/I) was a randomly occurring variable in the 10–12 stem segments evaluated for each genotype tested under either dry or wet conditions. When the node was adopted as the loading point, the stem was cleared of the leaf sheaths located below it but not the one that was inserted into it.

### 4.4. Calculations

The area moment of inertia, *I*, was calculated by using Formula (1) from biometric measures taken with a caliper.

In a three-point bending test with a mid-span load, the stem bending strength (*B_S_*) is calculated as the maximum bending moment at the point of maximum deflection [5,12,15]:*B_S_* = *F*_S_ · ℓ/4,(4)
in which *F*_S_ is the measured (ultimate) failure strength (that is, the maximum force that a stem portion can withstand while being bent), and ℓ is the span length of the stem segments laying between the supports (ℓ = 90 mm in our setup). The precision balance we used is actually a scale that measures weight and, as usual, it provides readings in units of mass, that is, grams. However, scales measure grams-force (gf), in fact. To obtain the failure strength (*F*_S_) expressed in newtons (N), the readings must be transformed in kilograms-force (kgf) and multiplied times 9.80665, since 1 kgf ≡ 9.80665 N.

Material strength (σ_b_) can, then, be calculated as the maximum material stress before failure [5] by rearranging Formula (3):σ_b_ = *B_S_* · *r*_e_/*I*.(5)

The modulus of elasticity (*E*) represents the characteristic stiffness of the material. In a three-point bending test, it can be calculated from the force (applied with a height gauge to measure deflection) that is necessary to bend a horizontal stem segment. The deflection of the stem (*h*) is [5]
*h* = *F*_b_ · ℓ^3^/(48 · *E* · *I*),(6)
where *F*_b_ is the applied bending force causing deflection *h*.

The slope of the linear regression of the measured values of *F*_b_ against *h* is ∆*F*_b_/∆*h*, and thus,
*F*_b_ = *h* · ∆*F*_b_/∆*h*,(7)
so that
∆*h*/∆*F*_b_ = ℓ^3^/(48 · *E* · *I*).(8)

The modulus of elasticity was then calculated from the stress–strain plot (actually, a force–deflection plot) using the equation of a simply supported beam with a concentrated load at the center, as [5,11,16]
*E* = ℓ^3^/(48 · *I*) · ∆*F*_b_/∆*h.*(9)

A linear regression (typically with an r^2^ in the 0.990–1.000 range) of the sequence of five stress force measures (excluding the first two positive values observed) made with the height gauge at the initial, straight-line portion of the stress–strain curve of each bending test was used to calculate the regression slope ∆*F*_b_/∆*h* [N/mm].

### 4.5. Statistical Analysis

A high variation in mechanical traits among stem replicates was observed even before analyzing the data. The relevance of this issue and its consequences on data elaboration are explained in Appendix B. Statistical analysis was generated using SAS^®^ software (version 9.4; SAS^®^ Studio release 3.8—SAS OnDemand for Academics; copyright © 2012–2020, SAS Institute Inc., Cary, NC, USA). Analysis of Variance (ANOVA) was performed with the GLIMMIX procedure according to a generalized linear mixed model. The model was estimated with restricted maximum likelihood and optimized with a dual quasi-Newton technique. The Kenward–Roger correction for the (prediction) standard errors and denominator degrees of freedom was applied. A study of residuals is provided in Appendix C. In 2021 (for green plants), the experiment was based on a nested factorial (as the three levels of ‘segment position’ were associated within each ‘stem’ level). In the other years, the experiment was based on a randomized block design, and the ‘block’ factor was specified as the subject effect with a compound symmetry covariance structure [24]. Since the value of the response variable in every elementary plot (i.e., at each combination of the ‘genotype’ factor with the ‘dry/wet stem’ and ‘loading point’ testing conditions) was modeled as the average of measurements of multiple stem segments (about ten for each genotype or, when present, for each level of the ‘genotype’×‘dry/wet stem’ interaction), all the random factors meant to be used in the F-tests were made explicit in the model to ensure that appropriate error terms, rather than the residual variance, were used. A lognormal distribution was assumed for residuals of the response variables, except for the modulus of elasticity (*E*). Heterogeneous variances among levels of both the ‘genotype’ factor and—when present—the ‘dry/wet stem’ factor were modeled with the R matrix. The Dunnett’s T3 test was used for multiple comparisons of means among genotypes, separately for each testing condition.

For the analysis of data from the 2021 experiment, the repeated measures ANOVA model (which is suitable to analyze related dependent variables that represent different measurements of an attribute on the same subject) was based on the fixed factors ‘genotype’, ‘segment position’, and their interaction. The ‘stem’ effect was the sole random factor; it accounted for the fact that three related measures were taken at different positions on the same subject (i.e., the same stem). For multiple comparisons of means among genotypes, a further Bonferroni correction (*p* ≤ 0.05/3) was applied to account for the total number of comparisons across the three levels of the ‘segment position’ factor.

In the analysis of data from the 2022 experiment, the ANOVA model was based on the fixed factors ‘genotype’, ‘dry/wet stem’, and their interaction. As the ‘dry/wet stem’ factor was applied within each genotype for each block (that is, ‘genotype’ was the whole plot factor), a split-plot design was modeled with ‘dry/wet stem’ as the subplot factor. Random factors included ‘block’, ‘block’×‘genotype’, and ‘block’×‘dry/wet stem’(‘genotype’). For multiple comparisons of means among genotypes, a further Bonferroni correction (*p* ≤ 0.05/2) was applied in order to account for the total number of comparisons across the two levels of the ‘dry/wet stem’ factor.

For the statistical analysis of the data of the 2023 experiment assessed using the height gauge (in 2024), the ANOVA model only included the fixed factor ‘genotype’.

Data from the 2023 experiment with manual assessment (in 2024) were analyzed with an ANOVA based on the full factorial of the fixed factors ‘genotype’, ‘dry/wet stem’, and ‘loading point’. As ‘loading point’ was an additional variable applied within the ‘dry/wet stem’ factor, a split–split-plot design was modeled, with ‘dry/wet stem’ as the subplot factor and ‘loading point’ as the sub-subplot factor. Random factors included ‘block’, ‘block’ × ‘genotype’, block’ × ‘dry/wet stem’ (‘genotype’), and ‘block’ × ‘loading point’ (‘dry/wet stem’ × ‘genotype’). A test of ‘simple effects’ was performed to assess the effect of ‘loading point’ at each level of ‘dry/wet stem’ by means of a partitioned analysis of the LS-means of the interaction ‘loading point’ × ‘dry/wet stem’. A Bonferroni correction (*p* ≤ 0.05/2) was applied to account for testing across the two levels of ‘dry/wet stem’. For multiple comparisons of means among genotypes, an additional Bonferroni correction (*p* ≤ 0.05/4) was applied to account for the total number of comparisons across the four levels of the ‘loading point’ × ‘dry/wet stem’ interaction.

After the statistical analysis, the means of variables with a lognormal distribution were back-transformed with the omega method [25] for presentation in the tables.

## 5. Conclusions

Bending strength (*B_S_*) and material strength (σ_b_) can be assessed by manually applying the loading force with a fingertip in a three-point bending test, thereby speeding up the test. This facilitates the search for barley genotypes with better mechanical properties associated with lodging resistance. In addition, we used this manual method to compare the results of tests made by loading the bending force either on the node or the internode. It was thus shown that the pulvinus (which forms a complex with the node) is the weak point for mechanical resistance to bending in ripe barley culms. We also confirmed that wet stems have much lower mechanical resistance than dry ones. Moreover, when the bending moment is greater than the stem bending strength, wet stems do not show an abrupt breakdown; instead, they display a gradual creeping failure at the pulvinus. These findings fill a gap in previous studies on barley lodging, improving our understanding of how the mechanical properties of ripe barley stems determine lodging resistance in this crop and, thus, enhancing our capability to identify genotypes with better traits linked to lodging resistance. Our results, therefore, suggest breeding opportunities to improve lodging resistance in barley but also highlight the need for further exploration of the underlying mechanisms.

## Figures and Tables

**Figure 1 plants-13-03172-f001:**
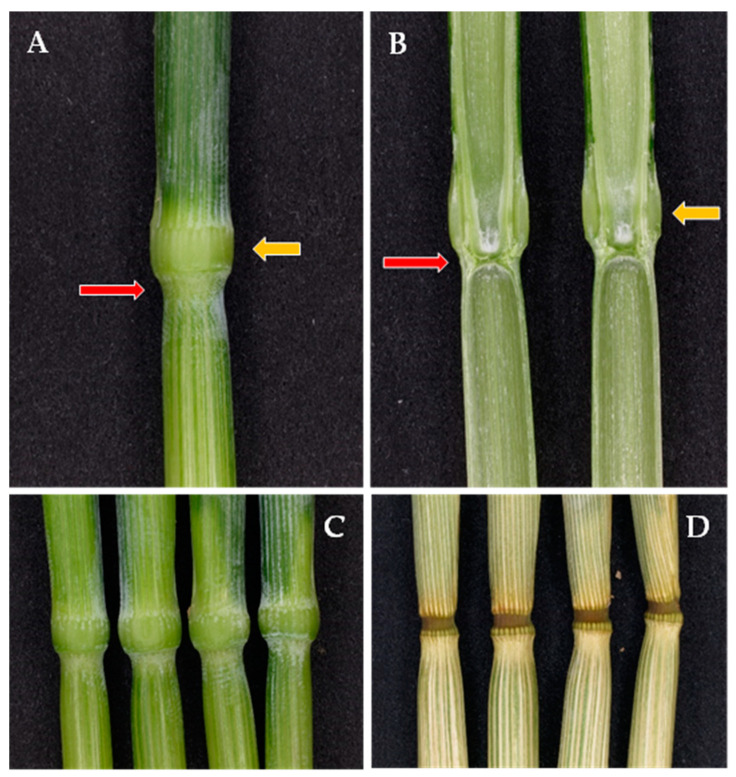
Features of the barley node–pulvinus complex: (**A**) The node–pulvinus complex comprises the node (red arrow), where the culm is solid as the transversal septa of adjacent internodes juxtapose, and the sheath pulvinus (yellow arrow), which is the swelling at the base of the leaf sheath, just above the node. (**B**) The two longitudinal sections of the same node–pulvinus complex. (**C**) Four green node–pulvinus complexes. (**D**) The same dried overnight at 105 °C.

**Figure 2 plants-13-03172-f002:**
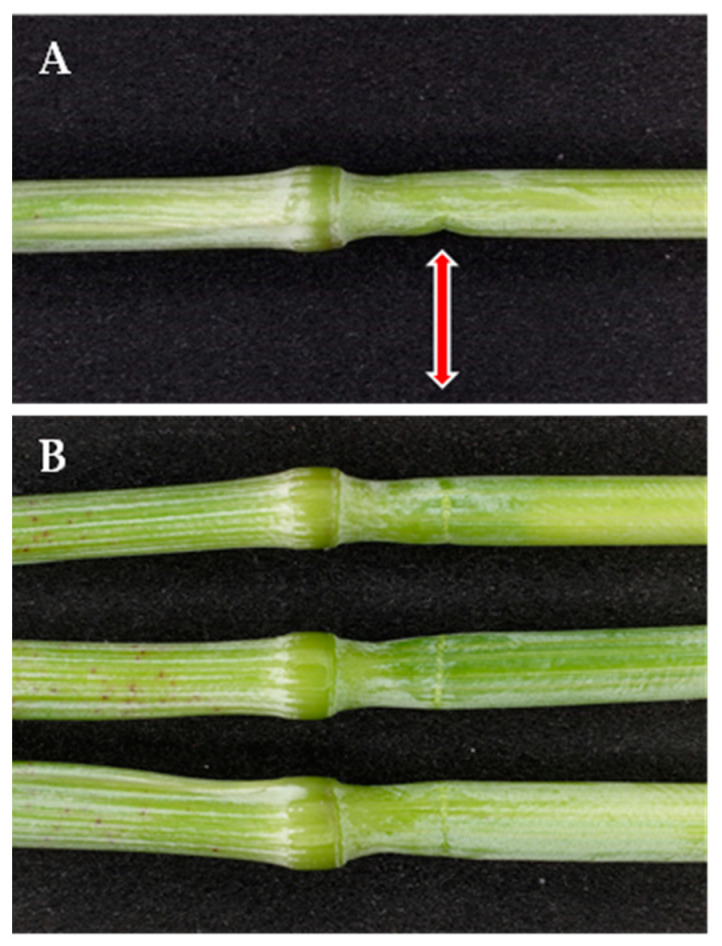
Buckling in three-point bending tests of green barley plants (cv Ketos). When the node is targeted as loading point in green stems (at soft dough stage of grain ripening) failure occurs at about 6 mm below the node (red arrow). (**A**) Lateral view of the kink formed at the site of stem failure; (**B**) viewed from above.

**Figure 3 plants-13-03172-f003:**
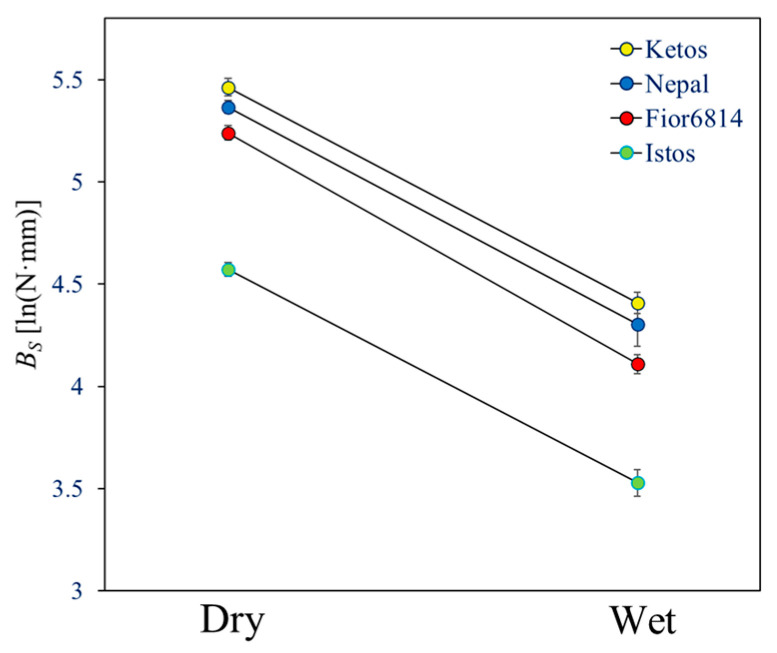
Bending strength (*B_S_*) across genotypes, assessed using the height gauge (2022 experiment). The averages of the four genotypes, tested as either dry or wet stems, are shown on the natural logarithm scale, where statistical significance of effects was established.

**Figure 4 plants-13-03172-f004:**
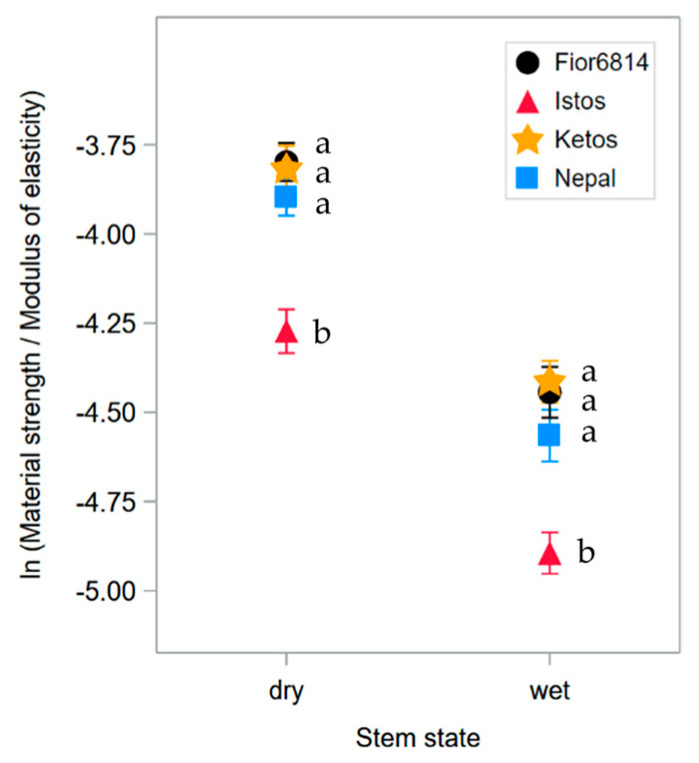
Ratio between σ_b_ and *E* of basal stem segments of ripe barley (2022 experiment), assessed with the height gauge. Averages of the natural logarithms of the ratio (that is, the values on the scale used to establish statistical significance of effects on σ_b_) are displayed. Genotypes are compared for both dry and wet stems. Genotypes with the same lower-case letter are not significantly different within each group of four genotypic means of either dry or wet stems (*p* ≤ 0.025; Dunnett’s T3 test).

**Figure 5 plants-13-03172-f005:**
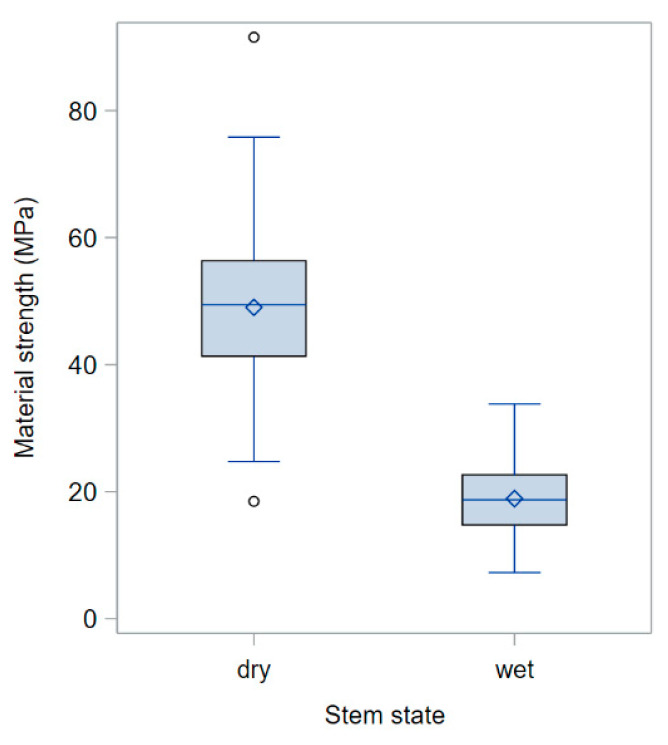
Overall averages of row data of material strength (σ_b_) for dry and wet stem segments (2023 experiment with manual assessment of ripe barley stems). The body of the box represents the interquartile range (IQR), that is, the range from the first (lower) quartile (Q1) to the third (upper) quartile (Q3) of the rank-ordered residuals (where the *k*th quartile is the score below which *k* quarts of the frequency distribution of residuals fall). The horizontal line inside each box indicates the median value (or second quartile, Q2), whereas the diamond represents the mean value. The whiskers that extend from each box indicate the minimum and maximum observed values that are outside of the IQR but within a distance ≤ 1.5·IQR (a conventional threshold beyond which data are considered outliers) either below the lower (Q1) or above the upper (Q3) edge of the box, respectively. Outliers (circles) are observations that are more extreme than 1.5·IQR, either below the Q1 edge or above the Q3 edge.

**Figure 6 plants-13-03172-f006:**
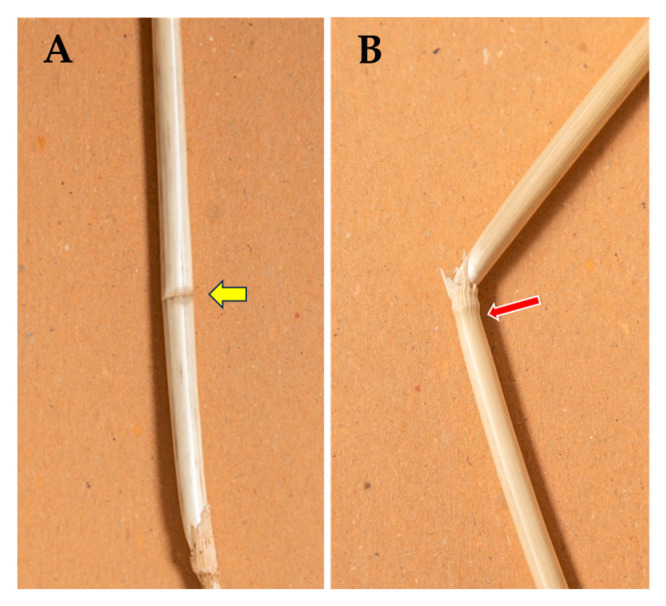
Buckling in three-point bending tests of dry ripe barley stems (line Fior6814). Failure (by manual application of the stress force) occurs in proximity to the loading point targeted for bending. (**A**) When the internode is targeted as loading point (about in the middle of the internode), failure occurs at the loading point, producing a typical kink of the stem (yellow arrow; viewed from above). (**B**) If the node is targeted as a loading point (red arrow), stem failure occurs at the pulvinus adjacent to the node (lateral view).

**Figure 7 plants-13-03172-f007:**
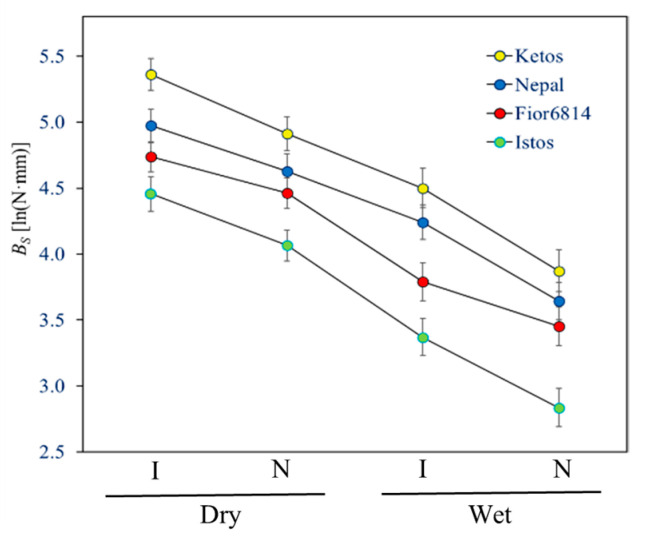
Assessment of bending strength (*B_S_*) under different conditions with the manual method. The averages of the four genotypes, tested at different conditions (dry/wet stems loaded at either the node, N, or internode, I), are shown on the natural logarithm scale, where statistical significance of effects was established.

**Figure 8 plants-13-03172-f008:**
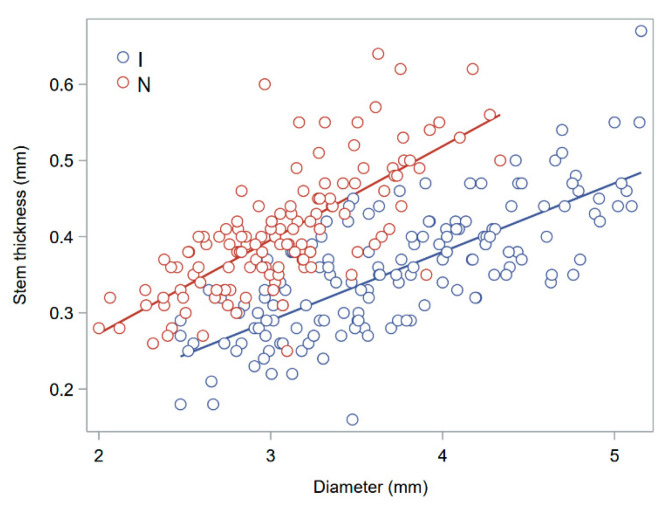
Relationship between stem wall thickness (*r*_e_ − *r*_i_) and stem diameter (*D*) in ripe barley. Red circles indicate that the internode was the loading point and blue circles indicate that the node was the loading point. A linear relationship is separately fitted for either loading points. Data from the manual assessment (2024) of the 2023 experimental samples.

**Figure 9 plants-13-03172-f009:**
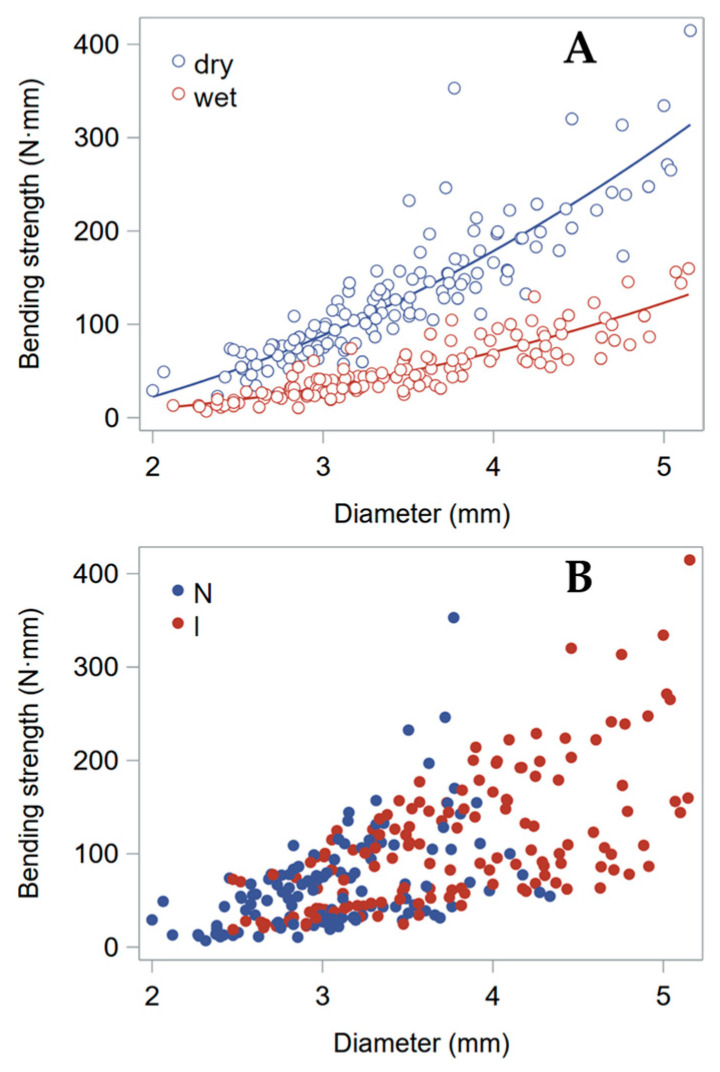
Relationship between the bending strength, *B_S_*, and the stem diameter, *D*: (**A**) Red circles represent wet stems, and blue circles represent dry stems. A quadratic relationship is separately fitted for the two stem moisture levels. (**B**) The same data are used to display the effect of the targeted loading point. Red filled circles indicate the loading force was applied at the internode, I, and blue circles indicate it was applied at the node, N. Data from the manual assessment (2024) of the 2023 experimental samples.

**Figure 10 plants-13-03172-f010:**
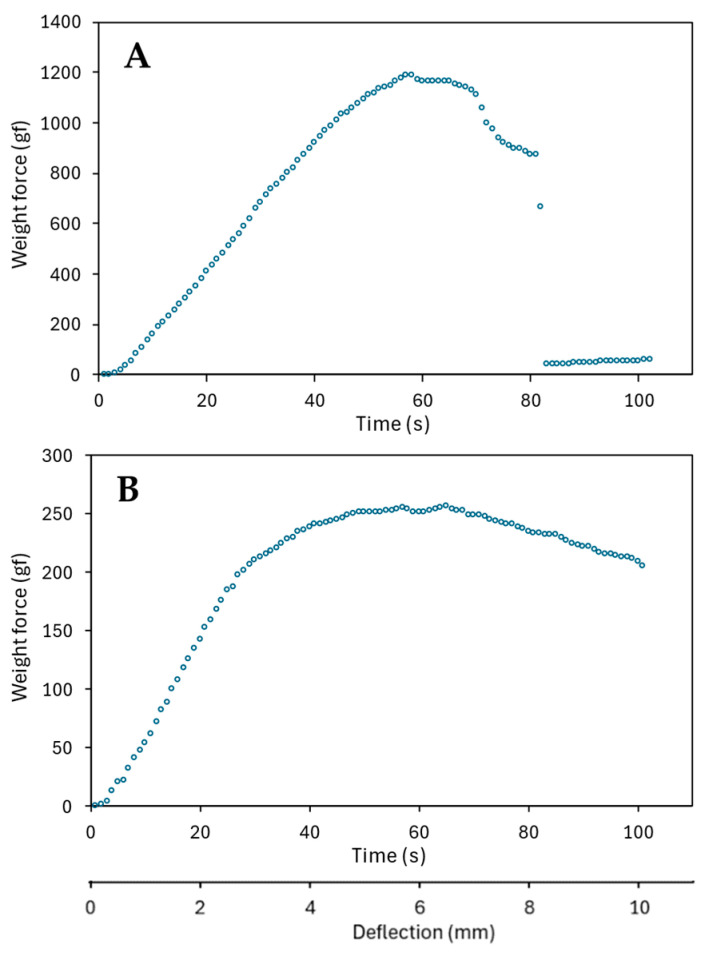
Representative force–deflection plots for ripe barley straw during the three-point bending test of (**A**) a dry stem segment and (**B**) a wet stem segment. As downward deflection was increased by 0.1 mm each second, the *x*-axis represents both a time and a strain scale, wherein 10 s corresponds to 1 mm of downward displacement at the loading point. Stems of Fior6814 (from the 2022 experiment) were used to realize these diagrams. The bending force was applied at the node with the height gauge.

**Figure 11 plants-13-03172-f011:**
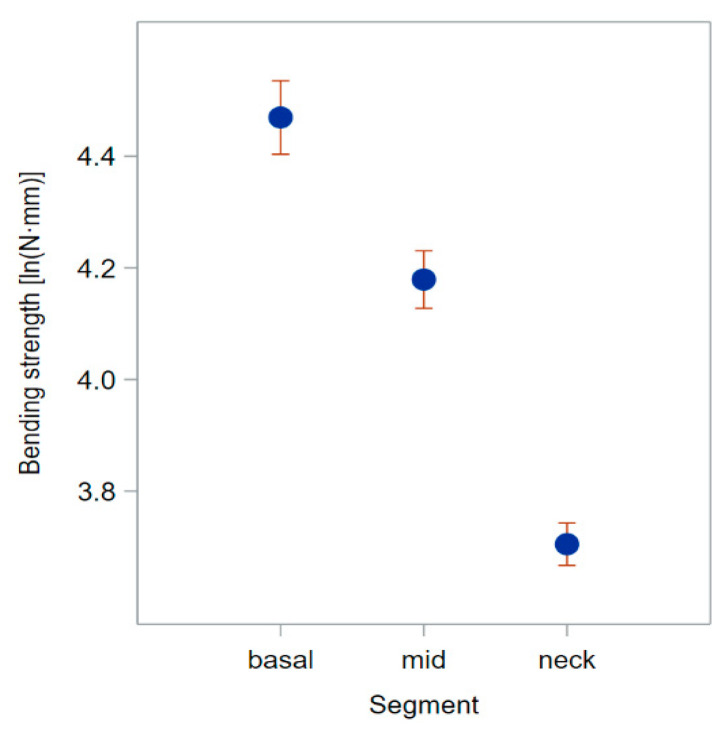
Bending strength (*B_S_*) of green plants (at early dough stage; 2021 experiment using the height gauge). The averages at the three positions at which stem segments were sampled are shown on the natural logarithm scale, where statistical significance of effects was established.

**Figure 12 plants-13-03172-f012:**
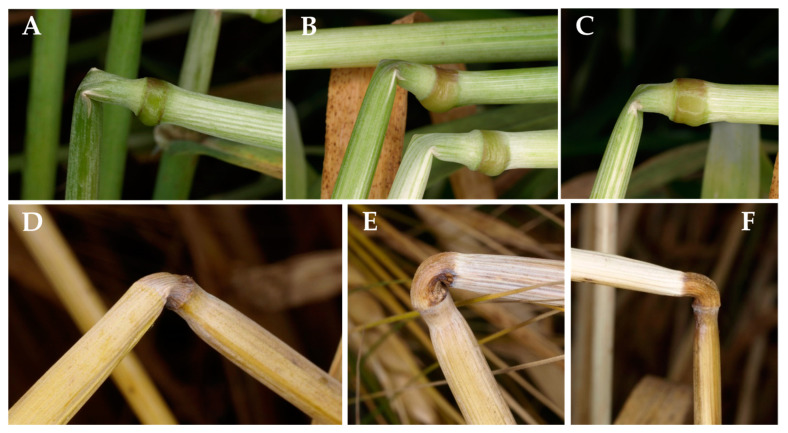
Failing point in barley brackling. When stem failure is observed in barley fields, it most often (though not always) occurs: (**A**–**C**) below the node, if the stem is green; (**D**–**F**) at the pulvinus, if the stem is ripe.

**Table 1 plants-13-03172-t001:** Mechanical parameters of stem segments from green plants (2021 experiment) assessed with the height gauge. Segments from different positions along the stem are compared over genotypes. For each parameter, significant differences between overall means of basal, mid, and neck stem segments are indicated with a different capital letter (*p* ≤ 0.05, Dunnett’s T3 test). Genotypes with the same lower-case letter are not significantly different within each group of four genotypic means (*p* ≤ 0.016; Dunnett’s T3 test).

Segment Position	Genotype	*D* (mm)	*r*_e_ − *r*_i_ (mm)	*I* (mm^4^)	*E* (GPa)	*EI* (mN·m^2^)	σ_b_ (MPa)	*B_S_* (N·mm)
Basal	Fior6814	4.00 ^a^	0.565 ^a^	9.71 ^a^	2.14 ^a^	20.1 ^a^	27.0 ^a^	132.0 ^a^
	Istos	3.47 ^b^	0.449 ^b^	5.23 ^b^	1.57 ^a^	7.4 ^b^	12.4 ^a^	33.4 ^b^
	Ketos	4.38 ^a^	0.571 ^a^	13.16 ^a^	1.71 ^a^	21.6 ^a^	19.7 ^a^	115.8 ^a^
	Tibet-A4	4.27 ^a^	0.607 ^a^	12.52 ^a^	2.06 ^a^	25.7 ^a^	24.3 ^a^	146.9 ^a^
	Mean	4.02 ^A^	0.544 ^A^	9.57 ^A^	1.87 ^A^	17.1 ^A^	20.0 ^A^	93.1 ^A^
Mid	Fior6814	3.82 ^b^	0.371 ^b^	6.35 ^b^	2.76 ^a^	17.6 ^a^	24.3 ^a^	80.4 ^b^
	Istos	3.61 ^b^	0.342 ^b^	4.99 ^b^	1.25 ^b^	5.5 ^b^	8.9 ^b^	21.9 ^c^
	Ketos	4.36 ^a^	0.446 ^a^	11.08 ^a^	1.76 ^b^	17.9 ^a^	20.0 ^a^	97.0 ^ab^
	Tibet-A4	4.51 ^a^	0.468 ^a^	12.84 ^a^	1.65 ^b^	20.9 ^a^	22.2 ^a^	129.0 ^a^
	Mean	4.06 ^A^	0.403 ^B^	8.20 ^A^	1.85 ^A^	13.8 ^B^	17.6 ^A^	68.5 ^B^
Neck	Fior6814	3.54 ^b^	0.320 ^b^	4.40 ^b^	1.77 ^a^	7.8 ^b^	15.2 ^a^	37.4 ^c^
	Istos	3.56 ^b^	0.307 ^b^	4.38 ^b^	0.68 ^c^	2.7 ^c^	7.9 ^b^	17.8 ^d^
	Ketos	4.21 ^a^	0.402 ^a^	9.12 ^a^	0.98 ^bc^	8.5 ^b^	13.8 ^a^	57.9 ^b^
	Tibet-A4	4.26 ^a^	0.429 ^a^	9.90 ^a^	1.22 ^b^	11.7 ^a^	17.5 ^a^	82.8 ^a^
	Mean	3.88 ^A^	0.361 ^C^	6.46 ^B^	1.16 ^B^	6.8 ^C^	13.1 ^B^	42.3 ^C^

**Table 2 plants-13-03172-t002:** Mechanical parameters of basal stem segments of ripe barley (from the 2022 experiment) assessed with the height gauge. Dry and wet stems are compared over genotypes. For each parameter, significant differences between overall means of wet vs dry stems are indicated by asterisks: * = *p* ≤ 0.05, ** = *p* ≤ 0.01, *** = *p* ≤ 0.001; ns = not significant (ANOVA). Genotypes with the same lower-case letter are not significantly different within each group of four genotypic means (*p* ≤ 0.025; Dunnett’s T3 test).

D/W	Genotype	*D* (mm)	*r*_e_ − *r*_i_ (mm)	*I* (mm^4^)	*E* (GPa)	*EI* (mN·m^2^)	σ_b_ (MPa)	*B_S_* (N·mm)
Dry	Fior6814	3.94 ^a^	0.486 ^b^	8.40 ^b^	2.20 ^b^	17.3 ^a^	47.4 ^a^	193.5 ^b^
	Istos	3.40 ^b^	0.412 ^c^	4.52 ^c^	2.81 ^a^	12.4 ^b^	38.4 ^b^	99.3 ^c^
	Ketos	4.15 ^a^	0.542 ^a^	10.89 ^a^	2.34 ^ab^	22.7 ^a^	49.4 ^a^	245.3 ^a^
	Tibet-A4	3.92 ^a^	0.535 ^a^	8.47 ^b^	2.60 ^ab^	21.2 ^a^	51.6 ^a^	219.5 ^ab^
	Mean	3.84 *	0.491 *	7.67 **	2.49 ***	17.9 ***	46.4 ***	179.3 ***
Wet	Fior6814	3.95 ^b^	0.477 ^b^	8.20 ^b^	1.39 ^a^	10.7 ^b^	15.5 ^a^	63.4 ^b^
	Istos	3.51 ^c^	0.452 ^b^	5.47 ^c^	1.62 ^a^	8.4 ^b^	11.9 ^b^	36.6 ^c^
	Ketos	4.35 ^a^	0.561 ^a^	12.60 ^a^	1.28 ^a^	15.2 ^a^	15.1 ^a^	86.1 ^a^
	Tibet-A4	4.09 ^ab^	0.544 ^a^	10.04 ^ab^	1.55 ^a^	15.0 ^a^	16.9 ^a^	82.6 ^ab^
	Mean	3.96 *	0.507 *	8.73 **	1.46 ***	11.9 ***	14.7 ***	63.7 ***

**Table 3 plants-13-03172-t003:** Pearson coefficients of linear correlation for dry stems (2022 experiment). Significance of correlation is indicated by asterisks: * = *p* ≤ 0.05, ** = *p* ≤ 0.01, *** = *p* ≤ 0.001; ns = not significant.

Parameter	*D* (mm)	*r*_e_ − *r*_i_ (mm)	*I* (mm^4^)	*E* (GPa)	*EI* (mN·m^2^)	σ_b_ (MPa)	*B_S_* (N·mm)
*D*	1	0.71 ***	0.96 ***	−0.60 ***	0.64 ***	0.12 ^ns^	0.86 ***
*r*_e_ − *r*_i_		1	0.75 ***	−0.36 ***	0.66 ***	0.20 *	0.76 ***
*I*			1	−0.58 ***	0.61 ***	0.06 ^ns^	0.84 ***
*E*				1	0.16 ^ns^	0.09 ^ns^	−0.44 ***
*E* *I*					1	0.28 **	0.67 ***
σ_b_						1	0.55 ***
*B_S_*							1

**Table 4 plants-13-03172-t004:** Mechanical features of ripe stems sampled in 2023. Values assessed on dry stem segments with the wedge-shaped tip of the height gauge anvil pointed at the node. Ten basal segments were assessed for each genotype. In each column, values with the same letter are not significantly different (significance at *p* ≤ 0.05, Dunnett’s T3 test).

Genotype	*D* (mm)	*r*_e_ − *r*_i_ (mm)	*I* (mm^4^)	*E* (GPa)	*EI* (mN·m^2^)	σ_b_ (MPa)	*B_S_* (N·mm)
Fior6814	3.05 ^b^	0.395 ^b^	3.09 ^b^	4.29 ^a^	12.9 ^b^	53.3 ^a^	105.9 ^a^
Istos	2.67 ^c^	0.321 ^c^	1.74 ^c^	4.47 ^a^	7.4 ^c^	41.5 ^a^	52.9 ^b^
Ketos	3.62 ^a^	0.443 ^ab^	5.90 ^a^	3.50 ^a^	20.1 ^a^	44.9 ^a^	143.5 ^a^
Tibet-A4	3.39 ^a^	0.476 ^a^	4.94 ^a^	4.11 ^a^	19.8 ^a^	45.0 ^a^	128.4 ^a^

**Table 5 plants-13-03172-t005:** Mechanical parameters of basal stem segments of ripe barley (from the 2023 experiment) with σ_b_ and *B_S_* assessed by manual application of stress force. Dry/wet (D/W) stems are compared over node/internode loading points (N/I) across genotypes. For each parameter, significant differences between overall means of wet vs dry stems are indicated by asterisks: *** = *p* ≤ 0.001; ns = not significant (ANOVA). Significant differences between means of measurements obtained pointing either the node or internode are indicated with a different capital letter, separately for dry and wet stems (*p* ≤ 0.025, ANOVA, ‘simple effect’ test). Genotypes with the same lower-case letter are not significantly different within each testing condition (*p* ≤ 0.0125, Dunnett’s T3 test).

D/W	N/I	Genotype	*D* (mm)	*r*_e_ − *r*_i_ (mm)	*I* (mm^4^)	σ_b_ (MPa) ^1^	*B_S_* (N·mm)
Dry	N	Fior6814	3.00 ^bc^	0.394 ^a^	2.88 ^bc^	47.6 ^a^	89.1 ^bc^
		Istos	2.67 ^c^	0.368 ^a^	1.93 ^c^	45.1 ^a^	61.2 ^c^
		Ketos	3.55 ^a^	0.479 ^a^	5.88 ^a^	45.3 ^a^	143.7 ^a^
		Tibet-A4	3.20 ^ab^	0.418 ^a^	3.71 ^ab^	48.0 ^a^	106.0 ^ab^
		Mean	3.09 ^B^	0.413 ^A^	3.32 ^B^	46.5 ^B^	95.5 ^B^
	I	Fior6814	3.41 ^bc^	0.302 ^b^	3.64 ^bc^	56.1 ^a^	117.1 ^bc^
		Istos	2.96 ^c^	0.307 ^b^	2.44 ^c^	58.3 ^a^	90.5 ^c^
		Ketos	4.32 ^a^	0.421 ^a^	10.51 ^a^	48.4 ^a^	225.7 ^a^
		Tibet-A4	3.87 ^ab^	0.365 ^ab^	6.33 ^ab^	47.5 ^a^	150.0 ^ab^
		Mean	3.60 ^A^	0.346 ^B^	4.93 ^A^	52.4 ^A^	137.6 ^A^
	Mean		3.34 ^ns^	0.378 ^ns^	4.04 ^ns^	49.3 ***	114.6 ***
Wet	N	Fior6814	3.12 ^a^	0.398 ^a^	3.42 ^ab^	16.4 ^a^	34.0 ^a^
		Istos	2.57 ^b^	0.347 ^a^	1.64 ^b^	14.9 ^a^	18.2 ^b^
		Ketos	3.60 ^a^	0.471 ^a^	6.44 ^a^	16.1 ^a^	52.9 ^a^
		Tibet-A4	3.27 ^a^	0.414 ^a^	4.04 ^a^	17.4 ^a^	40.3 ^a^
		Mean	3.12 ^B^	0.405 ^A^	3.47 ^B^	16.2 ^B^	33.9 ^B^
	I	Fior6814	3.32 ^b^	0.311 ^ab^	3.52 ^b^	23.6 ^a^	47.6 ^bc^
		Istos	2.85 ^b^	0.283 ^b^	2.00 ^b^	23.2 ^a^	31.1 ^c^
		Ketos	4.43 ^a^	0.414 ^a^	11.87 ^a^	20.2 ^a^	99.1 ^a^
		Tibet-A4	4.14 ^a^	0.369 ^ab^	8.25 ^a^	19.4 ^a^	73.2 ^a^
		Mean	3.63 ^A^	0.341 ^B^	5.13 ^A^	21.5 ^A^	57.3 ^A^
	Mean		3.36 ^ns^	0.371 ^ns^	4.22 ^ns^	18.7 ***	44.1 ***

^1^ Values of σ_b_ for tests wherein the node was targeted as loading point are displayed in red because they ought to be taken with caution, as explained in the text.

**Table 6 plants-13-03172-t006:** Characteristics of the genotypes used in this study.

Genotype	Status	Spike Type	Habitus	Height (cm)	Maturity	Lodging Resistance at Harvest Time (1–9 Scale)	Origin
Fior6814	breeding line	two-row	winter	77.3	medium	6	Italy
Ketos	commercial cultivar	six-row	winter	75.7	medium	7	Italy
Istos	old cultivar	two-row	winter	72.4	medium	3	France
Tibet-A4	traditional landrace	six-row	spring	136.5	early	1	Nepal

## Data Availability

Dataset available on request from the authors.

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
