# Peer review of "The Pulvinus Is the Weak Point for Stem Lodging Resistance in Ripe Barley"

_plants, 2024, doi:10.3390/plants13223172_

Round 1
Reviewer 1 Report
Comments and Suggestions for Authors
- Please improve your Abstract. Please use the following checklist to ensure you have included the necessary information: (1) Background and research question(s): 1-2 sentences; (2) Theoretical or conceptual framework; (3) Research Design and Methodology; (4) Results/Key findings: 3-5 sentences. (5) Implications of your study (1-2 sentences).
- Highlight the research gap in previous studies and clearly state the novelty of this research.
- The paper need completely re-arrangement from Introduction-Methods& Materials - Results - Discussion and Conclusion. This is the standard for a good article
- Figure 1 should be placed at the Materials and Methods section
- What is the motivation of this study?
- The contents of this manuscript can be streamlined as its present state is too wordy
- Where is the conclusion of your study???
- In the conclusion section, mention the implications of your research and its contributions to scientific knowledge. Relate your work to global interests and its worldwide importance to engage readers.
Comments on the Quality of English Language
Minor English checking
Author Response
Comment 1: - Please improve your Abstract. Please use the following checklist to ensure you have included the necessary information: (1) Background and research question(s): 1-2 sentences; (2) Theoretical or conceptual framework; (3) Research Design and Methodology; (4) Results/Key findings: 3-5 sentences. (5) Implications of your study (1-2 sentences).
Authors’ response: We have revised the abstract as recommended.
Comment 2: - Highlight the research gap in previous studies and clearly state the novelty of this research.
Authors’ response: This was done in the Conclusions. Thanks for your suggestion.
Comment 3: - The paper need completely re-arrangement from Introduction-Methods& Materials - Results - Discussion and Conclusion. This is the standard for a good article
Authors’ response: We have maintained the order of the sections as per the journal’s template.
Comment 4: - Figure 1 should be placed at the Materials and Methods section
Authors’ response: As, according to the journal’s template, the Materials and Methods section is placed after the Discussion, it would be pointless to place a figure that introduces basic knowledge about the studied topic, and is thus useful for understanding our experiments, at the end of the paper.
Comment 5: - What is the motivation of this study?
Authors’ response: We have briefly explained this point in the abstract.
Comment 6: - The contents of this manuscript can be streamlined as its present state is too wordy
Authors’ response: We have streamlined the contents of our manuscript.
Comment 7: - Where is the conclusion of your study???
Authors’ response: In the Instructions for Authors (https://www.mdpi.com/journal/plants/instructions) it is stated that the Conclusions section is not mandatory but can be added to the manuscript if the discussion is unusually long or complex. As the Reviewer seems to think that the discussion is unusually long, we have added the Conclusions section.
Comment 8: - In the conclusion section, mention the implications of your research and its contributions to scientific knowledge. Relate your work to global interests and its worldwide importance to engage readers.
Authors’ response: Thank you for this suggestion, we have added the recommended clarifications.
We appreciate the time you spent reviewing our manuscript and the constructive comments. We believe the revised version is much improved.
Reviewer 2 Report
Comments and Suggestions for Authors
This study offers valuable insights into the mechanical properties of barley stems and their relationship to lodging resistance, emphasizing key traits like stem diameter, thickness, stiffness, and strength. The use of a three-point bending test, with a manual method proposed as a reliable, resource-efficient alternative, provides practical value. Notably, the findings reveal that the pulvinus at the node is a weak point for bending resistance, with a significant drop in bending strength (BS) of 31% to 41% compared to the internode, and that wetting drastically reduces BS and material strength by 62%. These results highlight the need for further exploration of the underlying mechanisms and suggest breeding opportunities to enhance lodging resistance in barley genotypes.
Abstract
The objective of the study should be clearly stated, and the main findings need to be summarized concisely. Additionally, the logical flow of the abstract should be improved.
Figures 2 and 6
Specify which cultivar is being referred to in these figures.
Table 1
Clarify the meaning of the capital letters, either in the table title or in a footnote. Note that performing multiple comparisons for mean values in this way is statistically incorrect.
Figure 12
Explain what figures C-F are showing; further clarification is needed.
Figure 13
It is unnecessary to include a figure of the instrument in the main text, as it is commercially available.
Author Response
Comment 1: This study offers valuable insights into the mechanical properties of barley stems and their relationship to lodging resistance, emphasizing key traits like stem diameter, thickness, stiffness, and strength. The use of a three-point bending test, with a manual method proposed as a reliable, resource-efficient alternative, provides practical value. Notably, the findings reveal that the pulvinus at the node is a weak point for bending resistance, with a significant drop in bending strength (BS) of 31% to 41% compared to the internode, and that wetting drastically reduces BS and material strength by 62%. These results highlight the need for further exploration of the underlying mechanisms and suggest breeding opportunities to enhance lodging resistance in barley genotypes.
Authors’ response: thanks for the excellent summary of our work.
Comment 2: Abstract
The objective of the study should be clearly stated, and the main findings need to be summarized concisely. Additionally, the logical flow of the abstract should be improved.
Authors’ response: The abstract has been revised according to the Reviewer’s suggestions.
Comment 3: Figures 2 and 6
Specify which cultivar is being referred to in these figures.
Authors’ response: done.
Comment 4: Table 1
Clarify the meaning of the capital letters, either in the table title or in a footnote.
Authors’ response: in the table caption it is explained that “For each parameter, significant differences between overall means of basal, mid and neck stem segments are indicated with a different capital letter (p ≤ 0.05, Dunnett’s T3 test)”. We don’t think that rearranging this sentence can improve its clarity very much. So, please, point out what exactly is not clear in the meaning of this phrase.
Comment 5: Table 1
Note that performing multiple comparisons for mean values in this way is statistically incorrect.
Authors’ response: We don’t deem that performing multiple comparisons for mean values as shown in this table is statistically incorrect. Why does the Reviewer think it is?
Comment 6: Figure 12
Explain what figures C-F are showing; further clarification is needed.
Authors’ response: the Reviewer is fully right; inexplicability, we had completely messed up the capital letters referring to the panels in the legend of this Figure. Thank you for pointing out this silly mistake of ours.
Comment 7: Figure 13
It is unnecessary to include a figure of the instrument in the main text, as it is commercially available.
Authors’ response: Figure 13 has been deleted as requested.
We appreciate your time and effort, and we believe the revised version is much improved. Thank you very much for your valuable comments.
Reviewer 3 Report
Comments and Suggestions for Authors
The present work aims at identifying testing conditions that influence the assessment of stem mechanical properties, and, thus, the evaluation of genetic differences in lodging-related traits. Only 4 barley materials were analyzed and the three-point bending test is a common method to assess the mechanical properties of the stems, and there is no evidence to support the conclusion of the manuscript based on the field results. Thus, the manuscript should be reconsidered after major revisions.
Suggestions are as follows:
1. The manuscript is too long to reading. The text in introduction and discussion section should be shorten significantly.
2. The arrangements and main contents in the results section should be re-organized and optimized. It looks like an experimental summary rather than a research paper.
3. Considering that the total amount of data is not very large and the data from different years ultimately lead to same conclusion, so that the author may merge and display the data.
4. There were four barley materials of different genotype used for evaluation, are they sufficient to represent the characteristics of all barley?
5. Given the local climate conditions, strong winds during the mature stage often cause barley stem lodging. During the mature stage in the field, Is the stem failure occurs at the pulvinus when stem lodging? Please provide relevant statistical data.
Author Response
General comment: The present work aims at identifying testing conditions that influence the assessment of stem mechanical properties, and, thus, the evaluation of genetic differences in lodging-related traits. Only 4 barley materials were analyzed and the three-point bending test is a common method to assess the mechanical properties of the stems, and there is no evidence to support the conclusion of the manuscript based on the field results. Thus, the manuscript should be reconsidered after major revisions.
Authors’ response: as explained in the manuscript, the “field results” are preliminary observations aimed at showing that what was observed in the bending test matches with what happens in field conditions. Thus, no conclusions were based on such preliminary field results. All our conclusions are based on the bending tests. We could just remove any reference to these preliminary observations in the field and keep the conclusions as they are. However, even though scant, the field observations are consistent with the findings of the bending tests. We think that, though it has no statistical relevance, this consistency still is a useful piece of information, as suggested by Robertson et al. [10].
Regarding the number of barley materials analyzed, we chose four barley genotypes with largely different features of stem traits. As detailed below, in our response to the Reviewer’s comment 4, we think it was enough for the purpose of this study.
Suggestions are as follows:
Comment 1. The manuscript is too long to reading. The text in introduction and discussion section should be shorten significantly.
Authors’ response: we reduced the length of the text to improve its readability. However, please, note that for MDPI Plants “There is no restriction on the maximum length of the papers” (https://www.mdpi.com/journal/plants/about). We chose to publish in this journal also because there is no such restriction, and we therefore believe it is legitimate for us to submit a lengthy paper, insofar as the text is meaningful and relevant to our study.
Comment 2. The arrangements and main contents in the results section should be re-organized and optimized. It looks like an experimental summary rather than a research paper.
Authors’ response: we believe that the Results section clearly lays out our results, which include multiple experiments. Therefore, it is not clear to us how they should be re-organized and optimized. We would be grateful if the Reviewer could explain how the arrangements and main contents in the Results section should be re-organized and optimized.
Comment 3. Considering that the total amount of data is not very large and the data from different years ultimately lead to same conclusion, so that the author may merge and display the data.
Authors’ response: truly, the data from different years ultimately lead to the same conclusions, this is why we are confident about our findings. Unfortunately, they cannot be merged, because different experimental designs were applied across the years. On the one hand, this is a limitation of our study, due to the fact that, during the experiments, every year we realized that something was missing, and a novel idea was therefore introduced in the experiment. On the other hand, this is exactly how we reached our present conclusions.
Comment 4. There were four barley materials of different genotype used for evaluation, are they sufficient to represent the characteristics of all barley?
Authors’ response: we chose four barley genotypes with largely different features of stem traits. These different genotypes showed two common features: (1) their stems became less resistant to bending when wet, and (2) the ripe stems tended to fail at the pulvinus. As these barley genotypes are very different, phenotypically and genetically, we inferred that these should be common features of barleys. To what extent this affects barley crop cultivars world-wide or the whole Hordeum vulgare species, needs, we fully agree with the Reviewer, a much more extensive analysis. But this is a step further, beyond our present aim of “identifying testing conditions that influence the assessment of stem mechanical properties, and, thus, the evaluation of genetic differences in lodging-related traits, while also exploring potential improvements in this evaluation” (lines 215-217). Now that we know what to look at and how to test it, a more extensive study with many genotypes makes sense. We added some of the above-mentioned cautionary caveats at the end of the Discussion section.
Comment 5. Given the local climate conditions, strong winds during the mature stage often cause barley stem lodging. During the mature stage in the field, Is the stem failure occurs at the pulvinus when stem lodging? Please provide relevant statistical data.
Authors’ response: as explained in the text, it is very hard to make systematic observations on this aspect, because, apart from brackling, stem lodging commonly occurs at the base of the stem, below the soil surface. Like many other researchers before us, we tried to do systematic observations in the field but were unsuccessful in achieving clear results on this aspect. This is why we ultimately resort to the common three-point bending test to assess the mechanical properties of the stems. We are, therefore, unable to provide relevant statistical data on where and how exactly stem failure occurs in stem lodging in the field. The fact is that, for the same reasons, no study has ever provided “relevant statistical data” on this matter, as far as we know. If it were possible to provide relevant field statistical data on where and how exactly stem failure occurs in stem lodging, our study using the three-point bending test would have been mostly useless. Nevertheless, the preliminary field observations shown in Figure 12 represent, as said above, a useful check, in our opinion. As “Robertson et al. [10] recommended that the types and location of failure observed in the bending test should be similar to those found in the natural loading environment”, we provided preliminary information to support that this is indeed the case in our study.
Thank you for your sincere efforts, constructive comments, and raised points.
Round 2
Reviewer 3 Report
Comments and Suggestions for Authors
1. Although the author believes that long papers were allowed to publish in Plant, and that data from different years cannot be combined and the text can no longer be reduced, the reviewer insists that the manuscript should not be too long, especially when the data was not excessive, which is very unfriendly to the reader.
2. As in the manuscript, “The present work aims at identifying testing conditions that influence the assessment of stem mechanical properties”. So, the title could be modified to make it more representative of the content, and the abstract and conclusion, as well as in results and discussion sections, should focus on which “conditions” and parameters are appropriate for evaluating barley stem lodging resistance.
3. The author argued that “during the experiments, every year we realized that something was missing, and a novel idea was therefore introduced in the experiment”, actually, the main content of this manuscript compared the parameters of mechanical properties between green stems, mature culms (dry and wet), and mature stems with loading force on nodes and internodes, which were all measured by a three-point bending method. Reviewer suggest that “results section” really should be presented according to the “result” itself rather than years from which the data were collected. For example, the result of the 2021 data analysis would be better written as “Buckling of green barley plants”. Otherwise, you can't imagine how many pages it would take to explain the data collected from 10 years.
Author Response
- Although the author believes that long papers were allowed to publish in Plant, and that data from different years cannot be combined and the text can no longer be reduced, the reviewer insists that the manuscript should not be too long, especially when the data was not excessive, which is very unfriendly to the reader.
Authors’ response: We have made additional, minor cuts to the Discussion. However, please note that: (a) pages 26- 32 are Appendices; (b) the ‘Materials and Methods’ section is quite long (pages 21-25), but it is placed after the Discussion, so that a reader can read it just if she/he wishes to search for details of the study. (c) The Conclusions have been added because a Reviewer requested this section to be included. (d) We had already largely reduced the Introduction. (e) In the main text, there are 12 Figures (one has been deleted as requested by a Reviewer) and six tables; they occupy a noticeable part of the manuscript. Thus, we do not deem that the manuscript is exceedingly long. We acknowledge that a paper that is too long can be unfriendly to the reader, but even the omission of relevant information can be unfriendly to the reader. Although we respect the opinion of the Reviewer that appears to prioritize the former aspect, we give priority to the latter. If, however, the Reviewer deems that some parts of the manuscript are not meaningful or relevant enough to be included and wishes to point them out, we can reconsider them.
- As in the manuscript, “The present work aims at identifying testing conditions that influence the assessment of stem mechanical properties”. So, the title could be modified to make it more representative of the content, and the abstract and conclusion, as well as in results and discussion sections, should focus on which “conditions” and parameters are appropriate for evaluating barley stem lodging resistance.
Authors’ response: We think that there is a misunderstanding: in our manuscript we have never stated that some conditions are appropriate for evaluating barley stem lodging resistance whereas others are not. To the contrary, we wrote that “researchers can choose the best testing condition for their purposes: …” (line 545). We, rather, aimed “at identifying testing conditions that influence the assessment of stem mechanical properties” so as to understand why, up to now, the results of this assessment “have only a partial capability to predict lodging resistance” (line 11). We had tried to make this clear through lines 151-162, and we have now clarified it further by adding two sentences (lines 148-151). Moreover, since the sentence quoted by the Reviewer appears to cause some confusion, we reformulated it as: “The present work aims to identify variables that affect stem mechanical properties …” (lines 145-147). We hope our aim is now clearer and conveys an idea that is more consistent with the abstract, conclusion, as well as the results and discussion.
As for the title, we decided to stress the fact that the pulvinus is the weak point for bending resistance in ripe stems (but not in green ones), as this was a missing piece of information in the current understanding of lodging resistance. To make the title more representative of the whole content of the manuscript would mean to modify it as “An appraisal of factors that influence the mechanical properties of ripe barley plants shows that stems become less resistant to bending when wet and The Pulvinus Is the Weak Point for Stem Lodging Resistance” or alike. If the Academic Editor requests we change the title in this way, we will do it, but, in our opinion, this would make the title cumbersome. We are, nevertheless, open to suggestions about a better title.
- The author argued that “during the experiments, every year we realized that something was missing, and a novel idea was therefore introduced in the experiment”, actually, the main content of this manuscript compared the parameters of mechanical properties between green stems, mature culms (dry and wet), and mature stems with loading force on nodes and internodes, which were all measured by a three-point bending method. Reviewer suggest that “results section” really should be presented according to the “result” itself rather than years from which the data were collected. For example, the result of the 2021 data analysis would be better written as “Buckling of green barley plants”. Otherwise, you can't imagine how many pages it would take to explain the data collected from 10 years.
Authors’ response: We have modified the titles of the Results sections as recommended. In the previous round of revisions, we had not understood that this was the point criticized. Thank you for clarifying this aspect.
Round 3
Reviewer 3 Report
Comments and Suggestions for Authors
The author has made corresponding revisions to the entire text, and the reviewer recommends that the paper be accepted.